# ASSESSING EPISODIC MEMORY IN LLMS WITH SEQUENCE ORDER RECALL TASKS

## ABSTRACT

Current LLM benchmarks focus on evaluating models' memory of facts and semantic relations, primarily assessing semantic aspects of long-term memory. However, in humans, long-term memory also includes episodic memory, which links memories to their contexts, such as the time and place they occurred. The ability to contextualize memories is crucial for many cognitive tasks and everyday functions. Existing benchmarks have poor coverage of episodic memory. To address the gap in evaluating memory in LLMs, we define episodic memory for LLMs and introduce Sequence Order Recall Tasks (SORT), which we adapt from tasks used in cognitive psychology. SORT requires *causal* LLMs to recall the correct order of text segments, and provides a general framework that is both easily extendable and does not require any additional annotations. We present an initial evaluation dataset, Book-SORT, comprising 36k pairs of segments extracted from 9 books recently added to the public domain. Based on a human experiment with 155 participants, we show that humans can recall sequence order based on long-term memory of a book. We find that models can perform the task with high accuracy when relevant text is given in-context during the SORT evaluation. However, when presented with the book text only during training, LLMs' performance on SORT falls short. By evaluating a new aspect of memory, we believe that SORT will aid in the emerging development of memory-augmented models.

## 1 INTRODUCTION

Large language models (LLMs) have impressive performance on many benchmarks that test factual or semantic knowledge learned during training or in-context (Hendrycks et al., 2020; Ryo et al., 2023; Logan IV et al., 2019; Petroni et al., 2019; Yu et al., 2023; Sun et al., 2023). While these advances are noteworthy, the type of long-term knowledge that these datasets test is only one of several types that naturally intelligent systems store, retrieve, and update continuously over time (Norris, 2017; Izquierdo et al., 1999; McClelland et al., 1995). Current evaluation tasks do not assess episodic memory, which is a form of long-term knowledge thought to be important for cognitive function in humans and animals. Below we propose a definition of episodic memory in LLMs, which is based on the human literature (see Appendix A for a discussion).

---

**Definition 1.1: Episodic Memory in LLMs**

Episodic memory refers to knowledge in a language model that:

    (1)  is specific and unique to a particular sequence;

    (2)  is acquired through a single exposure to that sequence (single-shot learning);

    (3)  contains information about relations between parts (e.g. encountered events/items, incl. more abstract items) within that sequence;

    (4)  can still be retrieved when arbitrarily many tokens are processed in between encoding and retrieval;

    (5)  has functional implications, meaning the knowledge can be used by the model to answer explicit queries.

---

In contrast to semantic memory, episodic memory links memories to their contexts, such as the time and place they occurred. Research on human memory also originally focused on semantic, rather than episodic memory – however, researchers realized that one could distinguish the 'what' (semantic) content from the 'where' (spatial context) and 'when' (temporal context) Tulving (2002). This ability to organize memory based on spatial and temporal details enables us to reconstruct events that occurred in the possibly distant past, predict the future, and relate information across multiple events that are separated by time windows spanning a lifetime, capabilities crucial for many cognitive tasks and everyday functions. We propose SORT as a first benchmark to assess an important aspect of episodic memory.

The ability to link contextual details to stored information–particularly, details about temporal context– may be key to improving LLM performance on several tasks. More human-like episodic memory may improve models' continual learning and adaptation to shifting data distributions, performance on tasks requiring long contexts (e.g., long chat exchanges with a user), and source attribution via knowledge of where and when a memory was acquired, which could help to reduce or identify hallucinations.

To address the gap in evaluating crucial attributes of memory in causal LLMs, we propose the Sequence Order Recall Task (SORT), which we adapt from tasks in cognitive psychology that are used to assess long-term episodic memory in humans and animals (Eichenbaum, 2013; Davachi & DuBrow, 2015). Specifically, SORT requires a model to recall the correct order of sequential data, such as segments of text. We hope that SORT will be the first of many benchmarks that assess various aspects of episodic memory in LLMs.

We provide a specific instantiation of SORT that requires causal language models to recall the correct order of two segments sampled from text, along with a corresponding evaluation dataset–Book-SORT. Book-SORT contains over 36k pairs of text segments from 9 books, with variations in segment length (20 and 50 words) and distance between segments (up to 16k words). We chose books that were very recently released from U.S. copyright to minimize the possibility that LLMs were pre-trained on these texts. This allowed us to test three common methods of giving a causal language model access to a specific text: (1) during inference in-context, (2) during inference via retrieval augmented generation (RAG), and (3) during training via fine-tuning with a language modeling objective. Furthermore, we provide a human evaluation from 155 participants who had finished reading a whole book and were tested with no additional access to the book, showing that humans can recall segment order with up to 70% accuracy based on their long-term memory. While the ceiling performance on SORT is 100% (assuming that texts do not contain duplicate segments), our human data provides an important reference point to compare and contrast long-term memory across models and humans.

When given access to excerpts from the books in-context, we find that models achieve up to 95% accuracy with relevant 250-word excerpts but degrade quickly as longer excerpts are presented. Using Retrieval Augmented Generation, models can recall sequence order with limited performance. Finally, models fine-tuned with a language modeling objective on the book texts do not significantly improve their SORT performance, showing that parametric memory in current transformer models supports semantic but not episodic long-term memory.

Our main contributions can be summarized as follows:

- definition of episodic memory in the context of LLMs

- proposal of the self-supervised task SORT, which requires LLMs to recall the correct order of segments from a sequence and can be used to assess capabilities in causal LLMs that would be supported by episodic memory in humans

- a new dataset Book-SORT comprised of 36k samples from 9 public domain books and an evaluation framework that is easily extendable to new datasets

- first-of-its-kind human evaluation ($N = 155$) showing that humans are capable of recalling the order of text from an entire book based on long-term memory

- a comprehensive evaluation of open-source and closed language models on Book-SORT, showing that current models: i) have good in-context memory performance, when all necessary information is presented in the prompt and the prompt is short; ii) quickly lose the ability to recall sequence order as the excerpt provided in-context gets longer, though still far below their advertised context-lengths; (iii) fail to recall segment order based on

parametric memory formed via fine-tuning with a language modeling objective; (iv) perform worse on SORT with retrieval augmented memory than with in-context memory.

## 2 RELATED WORK

**Evaluation of parametric semantic memory in LLMs.** Benchmarks such as MMLU (Hendrycks et al., 2020), T-REx (Elsahar et al., 2018), LAMA (Petroni et al., 2019), WICE (Ryo et al., 2023), KoLA (Yu et al., 2023), and others (Sun et al., 2023) test models' retrieval and reasoning ability on different domains, such as recalling a chemistry fact.

Other benchmarks that partially evaluate LLM semantic memory are those that require reasoning using temporal (Ning et al., 2020; Zhou et al., 2021; Feng et al., 2023) (e.g. lunch happens before dinner), causal (Srivastava et al., 2023) (e.g. she is eating, therefore she is hungry), or other commonsense knowledge (e.g. food is edible) (Ismayilzada et al., 2023) acquired during pretraining. In contrast to these benchmarks, our work proposes a task that involves judgments regarding temporal context information about text segments that either (a) are available through in-context memory or (b) were otherwise previously presented to the model, e.g. via fine-tuning or Retrieval Augmented Generation.

**Evaluation of in-context memory in LLMs.** We evaluate in-context memory, in which the model has in-context access to all relevant text for the task. This relates to works that evaluate a model's ability to retrieve information from its context input, such as Needle In A Haystack (Kamradt, 2023) and FLenQA (Levy et al., 2024). These requirement 3 in our definition only minimally by testing the ability to retrieve an atomic piece of information regardless of its context.

Previous datasets and benchmarks that evaluate performance over long context lengths, such as Long Range Arena (Tay et al., 2021), SCROLLS (Shaham et al., 2022), and MULD (Hudson & Al Moubayed, 2022), are also relevant. The evaluation of in-context memory with SORT differs from these works by focusing on order information, which is key to episodic memory in humans. In NarrativeXL (Moskvichev & Mai, 2023) and NarrativeQA (Kočiský et al., 2017), models have to perform reading comprehension and free recall tasks when given entire books. SORT is different in that it places the focus on memory of segments where the complete context (all other segments) matters for the evaluation. This is not always the case in reading comprehension tasks where questions can be about atomic parts of the context that need to be retrieved.

**Tasks related to SORT.** Previously proposed tasks that most closely relate to SORT are BART's denoising training objective (Lewis et al., 2020), which permutes the order of sentences in a document and learns to reconstruct the correct order, and BERT's next sentence prediction objective (Devlin et al., 2019), which learns to predict whether two sentences follow each other in a text. SORT differs from these tasks, as it is not intended as a training objective, and it can include text segments with an arbitrary distance between each other in a document, possibly exceeding the context input length of the model. In ChapterBreak (Sun et al., 2022), long segments ending at a chapter boundary taken from a book are presented to an LLM along with multiple segments of chapter beginnings from the same book. The task for the LLM is then to tell which one is the directly following chapter and which are not. This suffix-identification task aims to evaluate narrative-understanding based reasoning about books, while we propose SORT as an evaluation for episodic memory in LLMs, involving both a model and a memory-insertion method. By evaluating a SORT baseline in which the models do not have access to relevant source texts, we show that memory is needed for SORT and general narrative-reasoning ability is not enough.

## 3 SEQUENCE ORDER RECALL TASK

We introduce a novel evaluation task: recalling the order of parts of a sequence, which we term the Sequence Order Recall Task (SORT). SORT is adapted from recency judgment tasks used in cognitive psychology to evaluate episodic memory in humans and animals (Eichenbaum, 2013; Davachi & DuBrow, 2015). In this task, a sequence is presented to a participant. Then, after some delay, the participant is asked to judge the order in which two segments of the sequence appeared. We adapt this task to test memory in models. The general task can be applied to any sequential domain, including video and audio. Here we focus on the text domain to evaluate LLMs (Fig. 1).

Sequence Order Recall Task (SORT)  Examples of methods to insert text-specific memory

**User:**
**<SORT-context>**
Your task is to read two segments taken from The Murder of Roger Ackroyd and recall in which order they appeared.

Segment A:
*Mademoiselle, it is of no importance what I think.*

Segment B:
*I saw the reflection of that thought in Poirot's next question.*

Question: Which segment, A or B, appeared first in The Murder of Roger Ackroyd?

**Assistant:**
Answer: Segment B ✅

**Baseline**

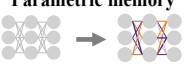

<SORT-context> is not given, performance depends on general knowledge and temporal order reasoning, since text-specific memory is not given.

**Parametric memory**

The model is fine-tuned on book-text with a language modeling objective. During Evaluation, <SORT-context> is not given, the model has to instead rely on memory in its weights.

**In-context memory**

A relevant excerpt from the source text containing both segments is presented in-context in place of <SORT-context>.

**Retrieval Augmented Memory**

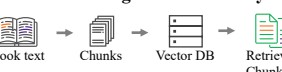
Book text    Chunks    Vector DB    Retrieved Chunks

Two chunks of text from the book are retrieved and presented in-context in place of <SORT-context>.

Figure 1: Overview of the Sequence Order Recall Task (SORT) to evaluate how models can access memory of temporal order. Left: Example task prompt for SORT. A prefix to the prompt can be given to assess in-context forms of memory. Right: Examples of methods to insert memory of specific texts into a model.

**Formal description of SORT.** The general form of the task can be described as follows. Let $\mathbf{X} \in \mathbb{R}^{T \times F}$ be sequential data, where $\mathbf{T}$ is the number of time-steps (e.g. token in a text) and $\mathbf{F}$ is the number of features (e.g. vocabulary size). We define start indices $\mathbf{t_j}$ and $\mathbf{t_k}$ for pairs of segments of length $\mathbf{L} \in \mathbb{N}^+$ in $\mathbf{X}$, such that both $\mathbf{t_j} < \mathbf{t_k}$ and $\mathbf{t_j} + \mathbf{L} \leq \mathbf{t_k}$. Using these, we extract non-overlapping segments from the original sequence $\mathbf{X}$ as $\widetilde{\mathbf{X}}_\mathbf{i} = \mathbf{X}[\mathbf{t_i} : \mathbf{t_i} + \mathbf{L} - \mathbf{1}, :]$. The order of segments $\widetilde{\mathbf{X}}_\mathbf{j}$ and $\widetilde{\mathbf{X}}_\mathbf{k}$ is randomized, yielding $[\widetilde{\mathbf{X}}_\mathbf{A} \ \widetilde{\mathbf{X}}_\mathbf{B}]$, which is then given as part of a model's input. The task for a model $\mathcal{M}_\theta$ is to infer whether $\mathbf{t_A} < \mathbf{t_B}$, i.e. in SORT, the task of a model is to predict which of two non-overlapping subsequences $\widetilde{\mathbf{X}}_\mathbf{A}$ and $\widetilde{\mathbf{X}}_\mathbf{B}$ has the lower starting index in $\mathbf{X}$. The task can be used to evaluate a variety of methods to include document-specific memory in models. To assess in-context memory, i.e. memory based on text presented in-context, the segments are preceded by $\mathbf{X}$ in the model's input. When assessing retrieval-augmented generation methods, instead of prepending $\mathbf{X}$, segments of $\mathbf{X}$ are retrieved and prepended. For the assessment of parametric long-term memory, $\mathbf{X}$ is not part of a model's input, instead the model's parameters (or a subset thereof) $\theta$ are a function of $\mathbf{X}$ via pre-training or fine-tuning: $\theta = f(\mathbf{X})$.

The general form of SORT is the following input, which can be preceded by additional context to insert a memory:

$$I_{SORT} = [P_{context} \ P_{task} \ P_{label_A} \ \widetilde{\mathbf{X}}_\mathbf{A} \ P_{label_B} \ \widetilde{\mathbf{X}}_\mathbf{B} \ P_{question} P_{answer}], \qquad (1)$$

where $\mathbf{P_{context}}$ can either be relevant context, such as (parts of) the source sequence $\mathbf{X}$ to assess in-context memory (stored in activation slots), or an empty string when parametric memory (stored in weights) is assessed; $\mathbf{P_{task}}$ instructs the model for the sequence order recall task to read two segments and describes the objective: answering which of the two labeled segments appears first in $\mathbf{X}$; $\mathbf{P_{label_A}}$ and $\mathbf{P_{label_B}}$ are the labels (e.g. the characters "A" and "B") for the first and second segment presented in the task $\widetilde{\mathbf{X}}_\mathbf{A}$ and $\widetilde{\mathbf{X}}_\mathbf{B}$; $\mathbf{P_{question}}$ repeats the SORT objective as a question; finally, $\mathbf{P_{answer}}$ provides the beginning of the answer string as "Answer: Segment".

### 3.1 EVALUATING LARGE LANGUAGE MODELS ON SORT

We greedily sample an answer token $\mathbf{a} = \mathbf{argmax}(\mathcal{M}_\theta(\mathbf{I}))$ from the model $\mathcal{M}_\theta$, which is parameterized by $\theta$, and decode the sampled answer token $\mathbf{a}$ as either "A" or "B".

The answer is evaluated as correct if it corresponds to the segment that truly appears first in $\mathbf{X}$. For proprietary (OpenAI) models that do not allow completing assistant responses with prepended text, we omit $\mathbf{P_{answer}}$. In this case we resort to generating a sequence of 25 tokens, and parse the generated text for A or B responses.

**Prompt selection.** Using a single prompt formulation across all models may bias the results. To prevent this, we compiled a set of 12 prompts that vary formulations in $\mathbf{P_{context}}$ and $\mathbf{P_{task}}$. For each

model, we evaluate each prompt on a held-out dataset of 400 samples and used the best performing prompt for each model. The full prompts and further details on prompt selection are given in Appendix C.2-C.3.

**Baseline without book-specific memory.** We want to ensure that performance on SORT is due to text-specific memory and not due to temporal order reasoning supported by more semantic forms of memory such as commonsense knowledge (e.g. lunch happens before dinner). We isolate the effects on SORT that are due to text-specific memory by contrasting performance between a baseline model that does not have access to the specific text and a model that has access to the sequences in one of various ways in which memory can be inserted.

## 3.2 INSERTING TEXT-SPECIFIC MEMORY INTO MODELS

We evaluate three examples of methods to insert text-specific memory into models: (1) via in-context presentation, (2) via fine-tuning with a language modeling objective, and (3) via retrieval augmented generation of short chunks of text in a book.

**In-context presentation.** When assessing in-context memory, $\mathbf{P_{context}}$ in Eq. 1 contains relevant excerpts from the source text along with the book title. The prompt includes the instruction to carefully read the text from the book (a list of used prompts is shown in Appendix 6). To test in-context memory, We make sure that excerpts contain both segments and vary the length of excerpts in our experiments.

**Finetuning with a language modeling objective.** Instead of presenting text from the books in the same prompt in which the SORT task is given, we are interested in parametric memory of the texts. In this condition, $\mathbf{P_{context}}$ in Eq. 1 is an empty string. To insert parametric memory of the source texts into a model, we fine-tune the model with a next-token prediction objective on the books, split into chunks of 5000 words and contextualized by the books' titles. Since we need to preserve the models' ability to understand and follow the task instructions (Allen-Zhu & Li, 2024), we fine-tune on a dataset which additionally includes 3,500 random instruction-following examples that are unrelated to SORT. This helps to prevent catastrophic forgetting during continued finetuning (Luo et al., 2024). We finetune on 8 A100 GPUs with an initial learning rate of 5e-6 and a batch size of 192. Full details of the fine-tuning setup are given in Appendix F and our code will be available. SORT should be informative about episodic memory more broadly, we did not train models on SORT (see Appendix F.4).

**Retrieval Augmented Generation.** To include memory of text via retrieval augmented generation (RAG), we built a typical naive RAG pipeline that relies on two separately pretrained models for the retriever and the reader (Gao et al., 2024). The retriever returns text passages from a database to serve as task context for the LLM (i.e. as $\mathbf{P_{context}}$, Eq. 1).

The retrieval database contained text embeddings of all passages from Book-SORT (Sec. 4). We used the LangChain recursive text splitter to chunk Book-SORT text into ∼1024 character, non-overlapping passages (average 183 words). Each passage was then encoded into a 1024-d vector using a high-performing, open-source text retrieval model (BGE-v1.5, (Xiao et al., 2024)). To retrieve the passages, we conduct an exact nearest neighbor search. The search returns the $k = 2$ nearest neighbors. We maintained this similarity order when inserting the retrieved passages into the prompt, i.e. the most similar passage appears first in $\mathbf{P_{context}}$.

## 4 BOOK-SORT DATASET AND EVALUATION

We created an English language dataset to evaluate episodic memory in humans and LLMs. The selected sequence data considered several factors: (1) we chose long texts (mean length = 72,700 words) that exceed the context windows of most transformer LLMs; (2) we used books to enhance memorability for human readers and facilitate our human evaluation experiment; (3) we selected books from *Project Gutenberg* that recently entered the U.S. public domain to avoid ethical and copyright issues, and minimize pre-training contamination in LLMs. Within these constraints, we aimed to maximize content diversity, including narrative fiction novels, a physics text, and an extended essay. Further details on the 9 books in the Book-SORT dataset are available in Appendix B.1.

### 4.1 BOOK-SORT CREATION

We constructed a dataset that varies across factors that can affect human or model performance on SORT. Based on prior reports on LLMs (Liu et al., 2024), we first varied (1) $L_E$, the length of the text excerpt presented in context. Since the typical standard context length of the LLMs in our study was 4096 tokens, we set $L_E = \{250, 1000, 2500\}$ words. For models with extended context windows, we also created datasets where $L_E = \{10000, 20000\}$ words, which excluded one book that was too short. Our pilot experiments on humans suggested two other factors that would affect task performance: (2) $L_S$, the length of the segments from the text, and (3) $D_S$, the distance between the segments in the original text. To mirror the human experiments, we set $L_S = \{20, 50\}$ words. We then created 4 different distance bins $D_S = \{d_0, d_1, d_2, d_3\}$, whose values were bounded by the excerpt length $L_E$ (Appendix Table 4).

Within each unique combination of the first two factors $L_E$ and $L_S$, we randomly sampled 110 excerpts from each of the 9 books (i.e. 100 samples for SORT evaluation, and 10 samples for prompt selection per book). All excerpts and segments began at a sentence boundary. Within each combination of $L_E, L_S$, we randomly sampled 4 different segment pairs, one from each distance bin $D_S$. This minimized the possibility that observing an effect of distance on SORT performance would be due to differences in the semantic content of the text segments. Finally, for all 110 trials within each of these 3 factors, we counterbalanced the correct answer. This yielded a well-controlled and easily extendable dataset of about $36K$ text segment pairs for SORT evaluation.

### 4.2 HUMAN LONG-TERM MEMORY EVALUATION

As a reference point (but not a performance ceiling), we further provide a human evaluation from 155 participants who had recently finished reading one of the 9 books in the Book-SORT dataset, *The Murder of Roger Ackroyd* (Christie, 1927). This evaluation assessed long-term memory, as the average time between reading and testing was 7.5 days, far surpassing short-term memory duration (Hasson et al., 2015). There is no previously reported data on long-term memory for entire books from large samples, so we designed an experiment to collect this data. Given the difficulty of recruiting participants to read lengthy books specifically for an experiment, we used a creative recruiting strategy: inviting members of the online reading community *Goodreads* who had recently finished *The Murder of Roger Ackroyd*. Participants completed an online survey within 30 days of finishing the book. The expected compensation for participation was $12 and the study was approved by the IRB at Anonymized University. We provide 1570 segment pair samples from 155 participants. Further details about this one-of-a-kind study are provided in Appendix B.3.

### 4.3 MODELS

We evaluate a selection of open models covering a broad range of scores on popular benchmarks such as MMLU (see Table 5) ranging from 7b to 8x22b parameter transformer models. Initial experiments with non-instruction-tuned models resulted in chance performance on Book-SORT (see Appendix E), which we attribute to the lack of instruction tuning[1], and thus focus on evaluating instruction-tuned models in this work. We have selected models from different model families including Llama3 (AI@Meta, 2024), Llama2 (Touvron et al., 2023), Mistral (Jiang et al., 2023), Mixtral (Jiang et al., 2024), Gemma (Team et al., 2024) and OpenAI GPTs (Achiam et al., 2023). For our experiments on finetuning as a method for inserting memory into models, we focus on two models Mistral-v0.2-7b-Instruct and Llama3-8b-Instruct because they allow full-parameter fine-tuning with 8 A100 GPUs.

## 5 RESULTS

We present empirical findings for a baseline without text-specific memory of the books in Book-SORT, as well as three methods to include memory, using 9 open-source models and 2 closed language models.

---

[1](Zhang et al., 2024b) provides an overview of instruction tuning approaches

Table 1: Baseline: SORT performance before models are exposed to the books in Book-SORT.

|  | Segment length 20 | Segment length 50 |
|---|---|---|
| Llama3-70b-inst | $0.52 \pm 0.007$ | $0.54 \pm 0.007$ |
| Llama3-8b-inst | $0.51 \pm 0.008$ | $0.52 \pm 0.007$ |
| Mixtral-8x22b-inst | $0.52 \pm 0.007$ | $0.55 \pm 0.007$ |
| Mixtral-8x7b-DPO-inst | $0.52 \pm 0.008$ | $0.54 \pm 0.008$ |
| Llama2-70b-inst | $0.51 \pm 0.007$ | $0.51 \pm 0.008$ |
| Gemma-1.1-7b-inst | $0.51 \pm 0.008$ | $0.51 \pm 0.007$ |
| Mistral-v0.2-7b-inst | $0.51 \pm 0.007$ | $0.51 \pm 0.008$ |
| Mistral-v0.1-7b-inst | $0.50 \pm 0.008$ | $0.50 \pm 0.008$ |
| Llama2-7b-inst | $0.50 \pm 0.008$ | $0.49 \pm 0.008$ |
| GPT-3.5-turbo | $0.52 \pm 0.009$ | $0.52 \pm 0.012$ |
| GPT-4 | $0.53 \pm 0.008$ | $0.57 \pm 0.007$ |

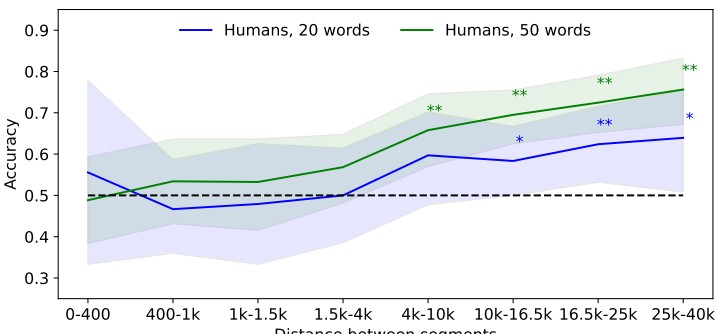

Figure 2: Human long-term memory performance on SORT for different segment lengths and distances between segments. Shaded areas depict bootstrapped 95% confidence intervals. Significant difference from chance is marked with asterisks ($^*$p-value$<0.05$,$^{**}$p-value$<0.01$).

## 5.1 BASELINE

**SORT requires memory specific to books in Book-SORT.** To validate that it is not possible to achieve high performance on Book-SORT without memory of the specific books that are included in the dataset, we evaluate models before they have access to the books. This shows that SORT requires memory of particular books and cannot be easily solved via temporal order reasoning (Hendrycks et al., 2020). We find that segment pairs with a very short and with a very long distance in the book allow a higher performance than chance (see Appendix D.1), indicating that some of these segment pairs can be ordered based not on memory but on reasoning or common-sense. However, none of the models have a high performance for any segment distance bin, as would be expected if SORT requires book-specific memory.

## 5.2 HUMAN EXPERIMENT

**Humans can perform in SORT based on long-term memory.** The results from human long-term memory (LTM) experiments, depicted in Figure 2, demonstrate that humans can perform in SORT based on long-term memory. The average accuracy is $0.64$ for segments of 50 words and $0.56$ for segments of 20 words). Human performance is higher for pairs of segments that have a greater distance in the book, with a peak accuracy of $0.76$ for distances greater than 25,000 words and 50-word segments. Binomial tests show that beyond a distance of 4000 words, humans perform statistically significantly better than chance. Note that we present these results as evidence that one possible information processing system–a human–can perform SORT based on long-term memory. Importantly, these results do not present the ceiling performance on the memory task that we propose. The expected ceiling performance on SORT is 100%, given that the books do not contain duplicated segments of text, which is less probable for longer segment lengths.

Table 2: Mean of in-context memory performance with 95% bootstrapped confidence interval. SORT-extend shows performance with excerpts of lengths 10000 and 20000 words, which exceeds most models' context lengths.

| Model name | Parameters | Max context | SORT | SORT-extend |
|---|---|---|---|---|
| Llama3-70b-inst | 70b | 8k | $0.92 \pm 0.020$ | / |
| Llama3-8b-inst | 8b | 8k | $0.93 \pm 0.007$ | / |
| Mixtral-8x22b-inst | 8x22b | 64k | $0.95 \pm 0.020$ | $0.79 \pm 0.038$ |
| Mixtral-8x7b-DPO-inst | 8x7b | 32k | $0.89 \pm 0.030$ | $0.56 \pm 0.058$ |
| Llama2-70b-inst | 70b | 8k | $0.77 \pm 0.040$ | / |
| Gemma-1.1-7b-inst | 7b | 8k | $0.85 \pm 0.010$ | / |
| Mistral-v0.2-7b-inst | 7b | 32k | $0.85 \pm 0.032$ | $0.65 \pm 0.045$ |
| Mistral-v0.1-7b-inst | 7b | 8k | $0.77 \pm 0.013$ | / |
| Llama2-7b-inst | 7b | 4k | $0.56 \pm 0.014$ | / |
| GPT-3.5-turbo | unknown | 16k | $0.86 \pm 0.010$ | / |

## 5.3 In-context memory

**Models generally perform well on SORT based on in-context memory.** Nearly all models achieve above 77% accuracy when given in-context access to relevant excerpts from the books, reaching up to 95% (Table 2). This indicates that very large models are not necessary to perform this task effectively, as demonstrated by the Llama3-8b model outperforming larger models such as Llama3-70b and Mixtral-8x7b-DPO.

**In-context memory performance increases with greater distance between segments.** We further evaluate the effect of another factor which may influence the model performance–the distance between the text segments in the excerpt. Figure 3b shows an increasing trend in accuracy as the distance between segments increases. This improvement in accuracy is consistent across excerpt lengths and is observed across all models (see Appendix D.2).

**In-context memory performance decreases with increasing excerpt length.** Average performance on longer excerpts (Table 2, SORT-extend) is substantially lower than in the standard context lengths, despite the presence of longer segment distances. For increasing excerpt lengths, we see a consistently monotonic decrease in average accuracy (Figures 3a and 3).

**Additional analyses.** Further analyses are presented in Appendix D.2. Models handle longer segments (50 words) slightly more effectively than shorter segments (20 words), with an improvement of up to $4\%$. We found no significant differences across books from different domains (Table 11-12).

## 5.4 Parametric Memory via Finetuning

**Full parameter fine-tuning on books with a language modeling objective did not improve SORT performance.** For Llama3-8b-Instruct and Mistral-7b-v0.2-Instruct, we do not observe any difference in performance on SORT after memory is inserted via fine-tuning on large chunks of book-text. A pairwise statistical analysis across epochs of fine-tuning, relative to two baselines that either exclude the books from the fine-tuning dataset or instead include only summaries of the books, shows no substantial improvement (see Appendix F).

## 5.5 Retrieval Augmented Memory

**RAG based memory leads to worse performance than in-context memory.** Due to the fact that the order of multiple passages from the same document is not preserved in a standard RAG setting, the performance is lower than in in-context memory and does not reach 70% accuracy for any distance between segments (Figure 4a). Curiously, we find that bigger models (i.e. Mixtral-8x22b-Instruct and Llama3-70b-Instruct) do not substantially outperform smaller models with RAG. Even when passages containing both segments are retrieved and presented in the correct order, we find that Llama3-8b-Instruct outperforms two much larger models on SORT (Figure 4b).

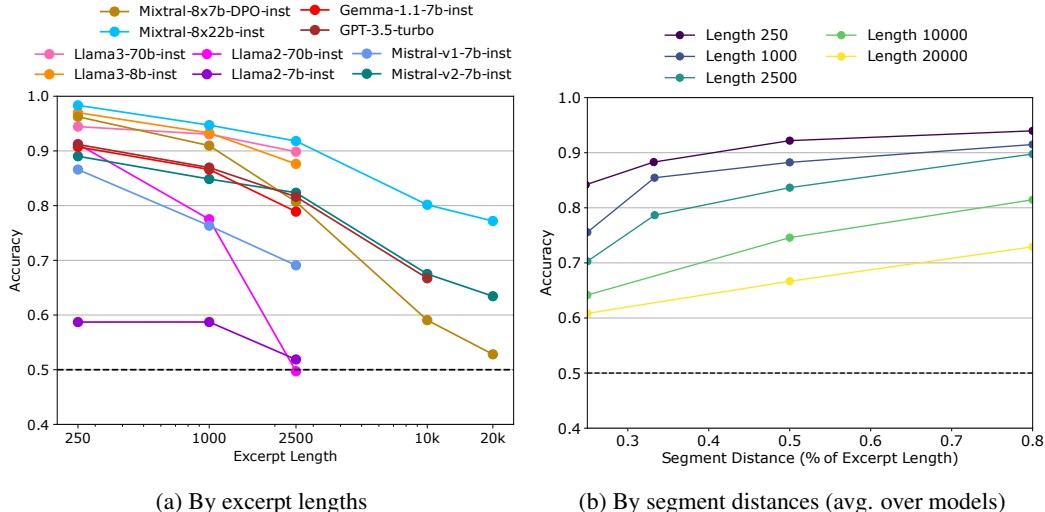

(a) By excerpt lengths       (b) By segment distances (avg. over models)

Figure 3: Factors affecting SORT performance based on in-context memory. (a) SORT accuracy by excerpt length. (b) Average over SORT performance of different models across segment distances for different excerpt lengths.

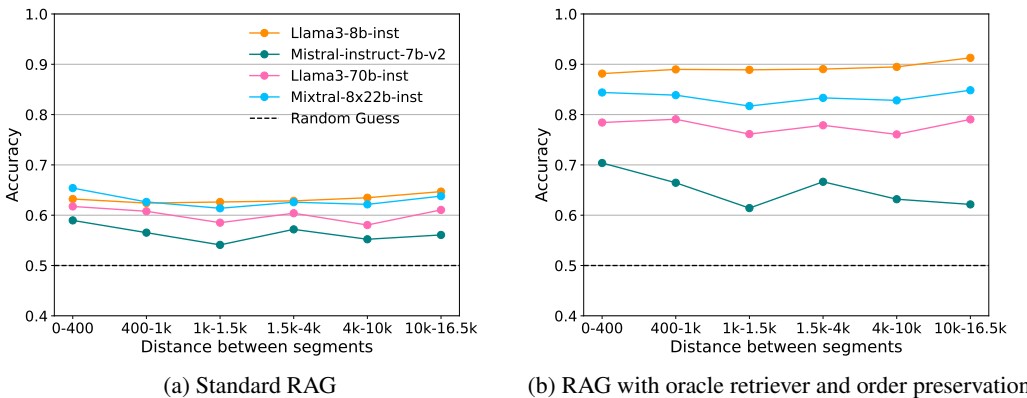

(a) Standard RAG       (b) RAG with oracle retriever and order preservation

Figure 4: SORT performance based on RAG memory. (a) Accuracy with standard RAG memory. (b) Accuracy with RAG memory given that the correct passages of text are retrieved and presented in the order in which they appeared in the books.

## 6 DISCUSSION

We provide a new evaluation task, SORT, for assessing episodic memory in causal large language models, that can be used with any text data and without the need for annotation. We created Book-SORT, a dataset for SORT based on books that were recently added to the public domain and we validated that book-specific memory is indeed needed to achieve high performance on Book-SORT. We evaluated three different ways to include memory of specific texts in a model to assess whether they support a key function of episodic memory. Below, we discuss our results for these methods in relation to episodic memory in humans.

**Is in-context memory a form of episodic memory?** Several links have been drawn between in-context memory in transformers and multiple models of episodic memory in humans (Ji-An et al., 2024; Whittington et al., 2022; 2024; Ellwood, 2024) and our results suggest that it does support sequence order recall. However, our in-context memory results suggest that performance degrades for sequence order recall, as it does for other tasks (Liu et al., 2024; Levy et al., 2024). We believe that the result of decreasing performance with more context supports a view of in-context memory as

an extended form of working memory. The two key problems with in-context memory that make it unlike long-term episodic memory are that it does not generalize well to arbitrarily long sequences and its cost increases as the context gets longer. In terms of our definition of episodic memory in LLMs (1), in-context memory disqualifies as supporting episodic memory because the fourth requirement is not fulfilled for current models.

**Is parametric memory in transformers a form of episodic memory?** High performance on benchmarks including MMLU suggests that parametric memory in LLMs learned via a language modeling objective can support semantic forms of memory (e.g. when recalling knowledge to answer factual questions). Our evaluation on SORT showing close to chance performance suggests that current forms of parametric memory insertion might not support functions similar to those of episodic memory.

**Is retrieval augmented memory a form of episodic memory?** Since it avoids the problems of context-length generalization, Retrieval Augmented Generation presents a potentially strong way to include memory of episodes via a retrieval process and subsequent in-context presentation. However, our results suggest that there is a lot of room for improvement over the performance of vanilla RAG. However, in vanilla RAG, retrieved segments are presented without surrounding context information (since chunks of the same document are independent). Order-preserving (OP) RAG (Yu et al., 2024) presents one way to retain relative positional information between retrieved passages as one kind of temporal context and can thereby increase performance on SORT. The episodic memory system in animals does not only bind temporal order information to memories but its context-binding generalizes to more abstract types of context (Eichenbaum, 2015a; Qiu et al., 2024) that would not be given in OP-RAG since memories are encoded independently of each other (i.e. the third criterion in our definition (1) is not properly fulfilled).

**Limitations.** Current high performing causal LLMs do not disclose their training data, which means that care needs to be taken in selecting suitable data to include in a SORT dataset. To minimize the probability that models have been trained on books used for our SORT evaluation, we curated Book-SORT based on books that were not publicly available when models were trained. However we cannot rule out that no copyrighted material was used in training of a model, which would require us to interpret results as indicating the effectiveness of additional rather than initial memory-insertion. Furthermore the reliance on instruction-following can limit the applicability to both non-instruction-tuned models and models that have poor instruction-following ability. While we provide a few examples of memory-insertion methods, we leave more extensive studies on how to induce episodic memories without relying on complete in-context presentation to future work.

**Future work.** Improving long-term memory in LLMs is an emerging area of research (Liu et al., 2023; Borgeaud et al., 2022; Fournier et al., 2023; Phang et al., 2023; Wang et al., 2024; Zhong et al., 2022; 2024), and SORT can be used to assess improvement in an crucial aspect of an important form of memory in new models. Specifically, improving episodic memory in models may improve models' continual learning, performance on tasks at long contexts such as extended chat exchanges with a user, and source attribution via knowledge of where and when a memory was acquired. Recent efforts have highlighted the potential of augmenting causal LLMs with additional episodic memory mechanisms (Fountas et al., 2024; Das et al., 2024), and we expect that SORT can be used to evaluate these classes of models, once such a model with a sufficiently strong instruction-following ability is released. Another possibility is to identify new and better methods to insert episodic memory of texts into existing models. Additionally, SORT can be extended to other types of inputs, such as audio and video, which can be used to evaluate episodic memory in multimodal models in the future.

**Conclusion.** The ability of LLMs to retain and retrieve long-term knowledge is crucial for their continued integration in many applications. Therefore, a more comprehensive and systematic evaluation of these abilities is needed. We believe that the new evaluation framework SORT offers a promising path for future research aimed at better understanding and improving these capabilities in foundation models.

**Ethics Statement.** To avoid ethical issues concerning copyright, we based Book-SORT on books that were recently added to the public domain. Our human experiment with 155 participants was approved by the IRB at Anonymized University and participants were compensated.

**Reproducibility Statement.** We will publicly release the Book-SORT dataset as well as all code to generate new SORT datasets and evaluate models on SORT. For open models, evaluation on Book-SORT is deterministic due to greedy sampling and the use of an answer prefix.

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

# A  DEFINING EPISODIC MEMORY FOR LLMs

Episodic memory is a new concept for LLMs that has not previously been explored in-depth. As such, it might be helpful to provide more explanations about the criteria that we adopted to define this form of memory in LLMs based on extensive research and mature theories about episodic memory in humans Tulving (2002); Andersen et al. (2006). Unlike human definitions of episodic memory, we do not require any notion of conscious personal experience, or a specific neural implementation (e.g. something like hippocampal dependence) Tulving (2002). Therefore this section will go through each criterion to provide additional background for our reasoning behind each item.

**(1) Episodic memory is specific to a single sequence.**  Episodic memory is a form of memory that is always specific to a single sequence and its unique temporal context. For humans, two experiences are easily distinguishable due to their high-dimensional nature, which makes it possible to find a number of features in which they differ. In the neuroscience literature, this is often described as memory representations that do not interfere with one another Sugar & Moser (2019); Colgin et al. (2008). This differentiates it from semantic memory, which can not necessarily be linked to a particular, unique sequence.

**(2) Episodic memory is learned through a single exposure to that sequence.**  Unlike semantic memory, episodic memories in humans and other animals can be based on single experiences, i.e. acquired through single-shot learning Liao & Losonczy (2024) or single-trial learning Schwartz & Evans (2001).

**(3) Episodic memory binds context to memory content.**  This is based on decades of research establishing that episodic memory binds the 'what', 'where', and 'when' of specific memories Sugar & Moser (2019). That is, episodic memories encode the spatial and temporal context associated with the encoded information, establishing a "cognitive map" in episodic memory O'Keefe & Nadel (1978). A contemporary theory of episodic memory posits that it is a more general "relational processing mechanism" Eichenbaum & Cohen (2014). It can include more abstract relations, e.g. functioning as a map of social space of memory (Eichenbaum, 2015b). This criterion does not just take the conservative view that episodic memory spans cognitive maps for space and time of events and items, but it can include more abstract relations between parts of a sequence, allowing for more abstractive relational mapping between and within parts of a sequence. The most conservative, well-studied, and easily testable aspects however remain temporal order and spatial memory, and we suggest that it is sensible to evaluate these types of contextual relations in LLMs before including more abstract relations, which we did not do in this work.

**(4) Episodic memory can potentially persist for an arbitrarily long time.**  In humans and other biological systems, episodic memory is specifically a form of long-term memory Squire & Zola (1996) that stores knowledge which can persist up to the span of a lifetime Mayes & Roberts (2001); Conway (2001). In LLMs this means episodic memory needs to allowing for (up to) arbitrarily many tokens in between the memory-sequence and a query about that sequence.

**(5) Episodic memory is generally accessible for explicit reasoning and communication.**  In classic views of human memory systems, episodic memory is a type of declarative, or explicit memory Squire & Zola (1996). This criterion is partly based on this classic view. While explicit memory is often characterized as consciously accessible memory, for LLMs, we map this to whether an LLM is able to answer explicit questions, suggesting that such information could explicitly be requested and then used in internal reasoning, which does not invoke any notion of consciousness. This is a *functionalist* criterion, as defined in the philosophy of mind Levin (2023). That is, our definition of episodic memory in deep learning models requires them to actually use the memory for general task solving. This criterion is also shared by neuroscientists studying episodic memory in non-human animals Hampton & Schwartz (2004).

# B  ADDITIONAL DETAILS ON BOOK-SORT DATA SET

**Preprocessing book text.** We wrote custom Python code to only retain the book text that formed a continuous narrative. We stripped the front and back matter of the book, and extracted chapter titles

if they existed. 8 of the 9 books contained individual section or chapter breaks. For these 8 books, we parsed the text corresponding to each chapter. Chapter titles or section headings (e.g. 'VI' to indicate section six) were removed, and all remaining text was concatenated. This string was split into words (assuming simple whitespace separators with Python `string.split()`) to produce a final text array for each book. This text array was sampled for the Book-SORT dataset.

## B.1 BOOK SELECTION

We provide details about the 9 books in Book-SORT in Table 3.

Table 3: Project Gutenberg metadata on Book-SORT books.

| ID | Title | Author | Word count | Release | Pub | LoCC* | Subjects |
|---|---|---|---|---|---|---|---|
| 69087 | The Murder of Roger Ackroyd | Christie, Agatha | 69,720 | 10/2/2022 | 1926 | PR | Detective and mystery stories; Fiction: Private investigators - England, Murder - Investigation, Belgians - England |
| 72578 | Tom Swift and His Talking Pictures | Appleton, Victor | 43,853 | 1/1/2024 | 1928 | PZ | Adventure stories; Motion pictures |
| 72600 | The Trumpeter of Krakow | Kelly, Eric Philbrook | 59,081 | 1/2/2024 | 1928 | PZ | Juvenile fiction: Middle Ages, Poland - History - Casimir IV, 1447-1492 |
| 72869 | Meet the Tiger | Charteris, Leslie | 79,946 | 2/4/2024 | 1928 | PR | Fiction: Private investigators - England; Detective and mystery stories |
| 72958 | Hunting for Hidden Gold | Dixon, Franklin W. | 42,354 | 2/14/2024 | 1928 | PZ | Juvenile fiction: Brothers, Gold mines and mining, Montana, Robbers and outlaws; Mystery and detective stories |
| 72963 | The Nature of the Physical World | Eddington, Arthur Stanley, Sir | 104,530 | 2/15/2024 | 1928 | Q | Physics - Philosophy; Science - Philosophy |
| 72972 | Money for Nothing | Wodehouse, P.G. (Pelham Grenville) | 82,331 | 2/16/2024 | 1928 | PR | Humorous stories; Fiction: Swindlers and swindling, Greed |
| 73017 | Pomona; or, the Future of English | De Selincourt, Basil | 9,273 | 2/22/2024 | 1928 | PE | English language |
| 73042 | The Well of Loneliness | Hall, Radclyffe | 163,217 | 2/26/2024 | 1928 | PR | Fiction: Lesbians - England - Social conditions |

*LoCC = Library of Congress classification.

## B.2 BETWEEN-SEGMENT DISTANCES

The segment distance $L_S$ for Book-SORT is sampled from one of four distance bins. The right edge of each bin is given in Table 4. Distance is computed between the beginning of the first segment and the beginning of the second segment. The minimum distance $L_S$ therefore produces adjacent, non-overlapping segments.

Table 4: Right edge of each distance bin used to create samples for Book-SORT.

| | Minimum | Bin0 | Bin1 | Bin2 | Bin3 |
|---|---|---|---|---|---|
| $L_E \leq 2,500$ | $L_S$ | $L_E/4$ | $L_E/3$ | $L_E/2$ | $L_E/0.8$ |
| $L_E \geq 10,000$ | $L_S$ | 1000 | $L_E/4$ | $L_E/2$ | $L_E/0.8$ |

## B.3 HUMAN STUDY DETAILS

**Participant compensation.** Participants were compensated via a lottery system with a chance to win a gift card to a popular book store. The expected value of the compensation came out to $12 per hour.

**Study design.** Each participant completed an online survey. First, the participant consented to the study, read a brief set of instructions, and completed a brief survey, including a question regarding when the participant finished reading the book. The complete set of survey questions is listed below. Each participant was then asked to answer "Which segment occurred first in the book?" for 10 randomly chosen text segment pairs from a total set of 540 unique segment pairs sampled from the whole book. We chose to present a sample number of trials to each participant to minimize interference effects from repeated memory retrieval (Kliegl & Bäuml, 2021). The presentation order of the text segments was randomized across participants. In the end, each participant was asked 4

simple questions about the book plot to verify that the participant had indeed read the book. Each participant was only allowed to participate in the study once.

**Demographics questions.**  The human participants were asked the following set of demographics questions before beginning the experiment:

1. I have finished the book The Murder of Roger Ackroyd [Options: True/False]

2. On what date did you finish the book? [Calendar question type]

3. Did you read or listen to the book? [Options: Read/Listen]

4. Was this your first time reading / listening to the book? [Options: Yes / No]

5. What is your age? [Options: 18-25, 25-35, 35-45, 45-55, 55-65, 65+]

6. What gender do you identify with? [Options: Female/Male/Other]

7. What is your experience with the English language? [Options: Native / Fluent / Advanced / Intermediate / Beginner]

8. How many books did you read or listen to in the past year? [Options: 1-2 / 3-5 / 6-10 / 10+]

We use the responses above to determine the number of days that have passed since finishing the book, and make this information available in the human dataset together with the responses.

**Inclusion criteria.**  We include data from participants who answered at least 3 of 4 plot questions correctly, and finished reading the book within 30 days of participating in the study. These inclusion criteria result in 155 participants.

## C  MODEL AND PROMPTING DETAILS

### C.1  MODEL DETAILS

We listed all models we used in this paper and their download links from HuggingFace in Table 5. For the OpenAI models, we used the gpt-3.5-turbo-0125 version of GPT-3.5, and gpt-4-turbo-2024-04-09 for GPT-4. Models were selected to cover a broad range of performance on more semantic/knowledge-based tasks such as those included in MMLU.

Table 5: Model Details

| Name in HuggingFace | Name in Paper | MMLU score |
|---|---|---|
| Llama-3-70B-Instruct | Llama3-70b-inst | 80.06 |
| Llama-3-8B-Instruct | Llama3-8b-inst | 66.60 |
| Mixtral-8x22B-Instruct-v0.1 | Mixtral-8x22b-inst | 77.77 |
| Nous-Hermes-2-Mixtral-8x7B-DPO | Mixtral-8x7b-DPO-inst | 72.28 |
| Mistral-7B-Instruct-v0.1 | Mistral-v1-7b-inst | 60.10 |
| Mistral-7B-Instruct-v0.2 | Mistral-v2-7b-inst | 60.07 |
| Llama-2-70b-chat | Llama2-70b-inst | 68.90 |
| Llama-2-7b-chat | Llama2-7b-inst | 45.30 |
| gemma-1.1-7b-inst | Gemma-1.1-7b-inst | 64.30 |

### C.2  PROMPTING

For our experiments with Book-SORT, we created a total of 12 prompts that are composed of two parts. The prompts differ in how they phrase the tasks. The first part contains instructions to read the text excerpt from the book as well as a placeholder for the actual excerpt. The second part of the prompt contains the description of SORT, including a mention of the book or document title as well as two segments from that document. We found that current open LLMs fail at the task even

with in-context access to the text, if they are asked to tell which segment appeared second or last. For this reason, we ran all experiments with the placeholder <position> set to "first". All of these prompts were preceded by the same generic system prompt: *"You are a helpful, respectful and honest assistant."*

Table 6: Selection of 12 prompts used for prompt validation

| No. | Reading instruction | SORT instruction |
|---|---|---|
| 1 | "Please take some time to thoroughly read and comprehend this extract from the book <booktitle>. The passage is as follows: <excerpt>" | "You will be shown pairs of text fragments from <booktitle>. Please select which of two fragments appeared <position> in the book. You will be shown 10 such pairs. <segments> Which fragment appeared <position> in the book, <label_0> or <label_1>?" |
| 2 | "I need you to thoroughly read and comprehend this extract from the book <booktitle>. The passage is as follows: <excerpt>" | "In this exercise, your objective is to identify the text segment, either <label_0> or <label_1>, that appeared <position> in <booktitle>. Please read the segments carefully to determine their order of appearance in <booktitle> and respond with either <label_0> or <label_1>: <segments> Which of these, <label_0> or <label_1>, was <position> in <booktitle>?" |
| 3 | "I need you to thoroughly read and comprehend this extract from the book <booktitle>. The passage is as follows: <excerpt>" | "Your task is to recall which text segment, either <label_0> or <label_1>, appeared <position> in the book <booktitle>. Please read the segments carefully to remember in which order they appeared in <booktitle> and respond with either <label_0> or <label_1>: <segments> Which of these, <label_0> or <label_1>, was <position> in the book <booktitle>?" |
| 4 | "I need you to thoroughly read and comprehend this extract from the book <booktitle>. The passage is as follows: <excerpt>" | "You will be shown two text segments, labeled as <label_0> and <label_1>. Please recall in which order they appeared in the book <booktitle> and tell me which one came <position>. Please read the segments carefully: <segments> Which of these two parts of the book, <label_0> or <label_1>, came <position> in the book <booktitle>?" |
| 5 | "I need you to thoroughly read and comprehend this extract from the book <booktitle>. The passage is as follows: <excerpt>" | "I will show you two short parts from a book, labeled as <label_0> or <label_1>. Your task is to tell me which of them appeared <position> in the book <booktitle>. Please read both segments carefully and try to remember where in the book they come from: <segments> Which of these, <label_0> or <label_1>, appeared <position> in the book <booktitle>?" |

Table 6: Selection of 13 prompts used for prompt validation

| No. | Reading instruction | SORT instruction |
|---|---|---|
| 6 | "I need you to thoroughly read and comprehend this extract from the book \<booktitle>. The passage is as follows: \<excerpt>" | "This is your task: Given two segments from a book, labeled as \<label_0> and \<label_1>, please tell me which of them appeared \<position> in \<booktitle>. Read both segments carefully and try to remember where in \<booktitle> they appeared: \<segments> Which of these, \<label_0> or \<label_1>, comes \<position> in the book \<booktitle>?" |
| 7 | "Please carefully read this excerpt from the book \<booktitle>. This is the relevant passage: \<excerpt>" | "You will be shown pairs of text fragments from \<booktitle>. Please select which of two fragments appeared \<position> in the book. You will be shown 10 such pairs. \<segments> Which fragment appeared \<position> in the book, \<label_0> or \<label_1>?" |
| 8 | "Please carefully read this excerpt from the book \<booktitle>. This is the relevant passage: \<excerpt>" | "In this exercise, your objective is to identify the text segment, either \<label_0> or \<label_1>, that appeared \<position> in \<booktitle>. Please read the segments carefully to determine their order of appearance in \<booktitle> and respond with either \<label_0> or \<label_1>: \<segments> Which of these, \<label_0> or \<label_1>, was \<position> in \<booktitle>?" |
| 9 | "Please carefully read this excerpt from the book \<booktitle>. This is the relevant passage: \<excerpt>" | "Your task is to recall which text segment, either \<label_0> or \<label_1>, appeared \<position> in the book \<booktitle>. Please read the segments carefully to remember in which order they appeared in \<booktitle> and respond with either \<label_0> or \<label_1>: \<segments> Which of these, \<label_0> or \<label_1>, was \<position> in the book \<booktitle>?" |
| 10 | "Please carefully read this excerpt from the book \<booktitle>. This is the relevant passage: \<excerpt>" | "You will be shown two text segments, labeled as \<label_0> and \<label_1>. Please recall in which order they appeared in the book \<booktitle> and tell me which one came \<position>. Please read the segments carefully: \<segments> Which of these two parts of the book, \<label_0> or \<label_1>, came \<position> in the book \<booktitle>?" |
| 11 | "Please carefully read this excerpt from the book \<booktitle>. This is the relevant passage: \<excerpt>" | "I will show you two short parts from a book, labeled as \<label_0> and \<label_1>. Your task is to tell me which of them appeared \<position> in the book \<booktitle>. Please read both segments carefully and try to remember where in the book they come from: \<segments> Which of these, \<label_0> or \<label_1>, appeared \<position> in the book \<booktitle>?" |

Table 6: Selection of 13 prompts used for prompt validation

| No. | Reading instruction | SORT instruction |
|-----|---------------------|------------------|
| 12 | "Please carefully read this excerpt from the book <booktitle>. This is the relevant passage: <excerpt>" | "This is your task: Given two segments from a book, labeled as <label_0> and <label_1>, please tell me which of them appeared <position> in <booktitle>. Read both segments carefully and try to remember where in <booktitle> they appeared: <segments> Which of these, <label_0> or <label_1>, comes <position> in the book <booktitle>?" |

### C.3 PER-MODEL RESULTS ON PROMPT SELECTION SWEEP

To identify the prompts that work best for each model, we take 400 segment-pair samples that we excluded from the main evaluation and evaluate models' in-context memory with all prompts shown in Table 6. To select the best prompt we considered both the proportion of A and B responses, which should be around 0.5, and the accuracy. We report the best selected prompts in Table 10 with numbers referring to the prompts presented in Table 6.

Table 7: Selected prompts for each model.

| Model Name | Best Prompt |
|------------|-------------|
| Llama3-70b-inst | 4 |
| Llama3-8b-inst | 3 |
| Mixtral-8x22b-inst | 4 |
| Llama2-70b-inst | 7 |
| Gemma-1.1-7b-inst | 8 |
| Mistral-v0.2-7b-inst | 3 |
| Mistral-v0.1-7b-inst | 2 |
| Llama2-7b-inst | 10 |
| GPT-3.5-turbo | 7 |
| GPT-4 | 7 |

### C.4 RAG PROMPT SELECTION

There were two different prompts to select for the retrieval-augmented generation experiments: the retrieval prompt (i.e. the search query), and the LLM prompt.

#### C.4.1 RETRIEVAL PROMPT (SEARCH QUERY)

The goal of retrieval in our RAG experiments is to find the text passages that will provide the most information about the segments for the sequence ordering task. After we created the vector database of all the text passages from Book-SORT, we formulated several different search queries (Table 8). We then ran retrieval using a validation subset of Book-SORT (50-word segments, 250-word excerpts from all books). The retrieval used the same database and text embedding model as described in the RAG portion of Section 3.2. The best search query was simple and only consisted of the segment text (query 8, Table 8). This search query is used for all RAG experiments.

#### C.4.2 RAG LLM PROMPTS

We followed a procedure similar to the one outlined in Section C.2. We created a total of 10 modifications to the reading instructions from Table 6.

Table 8: The search queries for the RAG experiment and their average retrieval recall@10 on a validation subset of Book-SORT (250 word excerpts, 50 word segments).

| No. | Search Query Text | Recall@10 |
|---|---|---|
| 0 | "Please determine the order in which the following text segments appeared in <booktitle>: <segments>" | 0.728 |
| 1 | "We need to put text segments from <booktitle> in order. These are the segments: <segments>" | 0.817 |
| 2 | "Please find these text segments from <booktitle>: <segments>" | 0.869 |
| 3 | "Please find these text segments from <booktitle> to provide context for the next task: <segments>" | 0.875 |
| 4 | "Which text chunks from <booktitle> contain the following segments? <segments>" | 0.802 |
| 5 | "Which text excerpts from <booktitle> contain the following segments? <segments>" | 0.799 |
| 6 | "Which text chunks from <booktitle> overlap with these text segments: <segments>" | 0.782 |
| 7 | "<booktitle> contains this text: <segments>" | 0.865 |
| 8 | "<segments>" | 0.906 |
| 9 | "<booktitle> <segments>" | 0.858 |

Table 9: RAG prompt modifications.

| No. | RAG Reading Instruction |
|---|---|
| 0 | "Here are some relevant excerpts from the book <booktitle>: <context>" |
| 1 | "The following excerpts from the book <booktitle>may be helpful context for the task. Context: <context>" |
| 2 | "Context: <context>" |
| 3 | "Searching a book database found these relevant text snippets: <context>" |
| 4 | "The following search results may be useful context: <context>" |
| 5 | "I will show you some relevant text found by searching a database of books: <context>" |
| 6 | "Please read some text deemed relevant for the task before performing the task. Relevant text: <context>" |
| 7 | "Please read these search results carefully to help you perform the task. Search results: <context>" |
| 8 | "Your objective may become easier with the use of these search results: <context>" |
| 9 | "This context may be helpful: <context>" |

### C.4.3 PER-MODEL RESULTS ON RAG PROMPT SELECTION

For a given LLM, we modified the reading instruction of the best prompt from Table 10 with each of the 10 options in Table 9. We then ran a sweep over the same 400 segment-pair samples detailed in Section C.3 and found the instruction that resulted in the highest performance on this held-out dataset.

Table 10: Best RAG instruction prompts for each model.

| Model Name | Best RAG Instruction No. |
|---|---|
| Llama3-70b-inst | 7 |
| Llama3-8b-inst | 7 |
| Mixtral-8x22b-inst | 3 |
| Mistral-v0.2-7b-inst | 6 |

# D  ADDITIONAL DETAILS ON BOOK-SORT  RESULTS

## D.1  MEMORY-LESS BASELINE RESULTS

Figure 5 shows performance on Book-SORT without any memory-insertion of the books used in Book-SORT. We find that performance is higher in segment pairs that are very proximal or very distant in the book, indicating that it might be easier to sort these pairs based on temporal order reasoning. Performance without additional memory-insertion is generally low, showing that memory is needed for SORT.

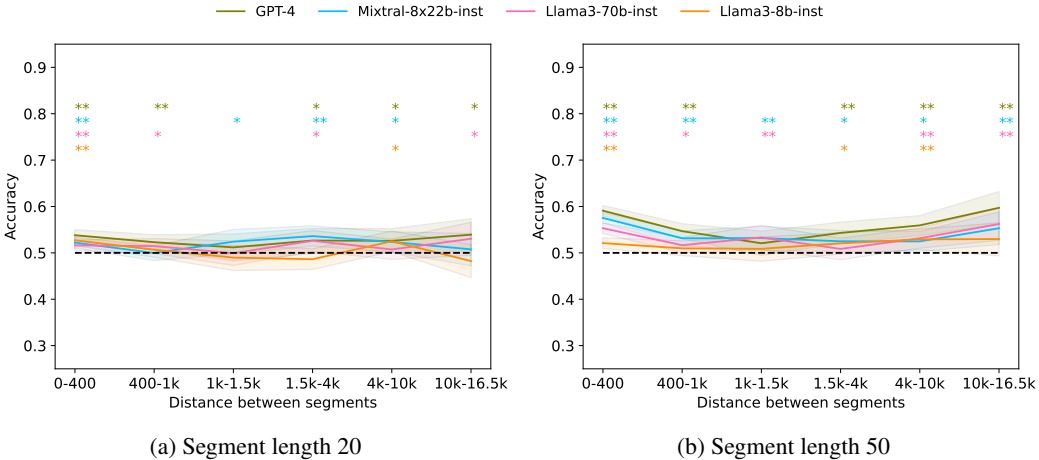

(a) Segment length 20            (b) Segment length 50

Figure 5: Baseline SORT performance without memory of books in Book-SORT. Significant difference from chance is marked with asterisks ($^*$p-value$<0.05$,$^{**}$p-value$<0.01$).

## D.2  IN-CONTEXT MEMORY FULL RESULTS

In this section, we provide a comprehensive overview of the in-context memory results across various models in Table 11 and Table 12. The table below illustrates the accuracy of different models on multiple books at segment lengths of 20 and 50 words. We observe that, while models generally perform slightly better with longer segments (50 words) compared to shorter ones (20 words), the improvement is modest, averaging up to $4\%$.

Table 11: Accuracy and Difference of Various Models on Multiple Books at Excerpt Lengths of 20 and 50, with in-context memory (Part 1)

| Model name | Book | SORT S20 | SORT S50 | SORT-Extend S20 | SORT-Extend S50 |
|---|---|---|---|---|---|
| Llama3-8b-inst | 69087 | 0.89±0.03 | 0.92±0.03 | / | / |
| Llama3-8b-inst | 72578 | 0.91±0.02 | 0.93±0.02 | / | / |
| Llama3-8b-inst | 72600 | 0.92±0.03 | 0.94±0.02 | / | / |
| Llama3-8b-inst | 72869 | 0.92±0.03 | 0.94±0.02 | / | / |
| Llama3-8b-inst | 72958 | 0.92±0.02 | 0.94±0.02 | / | / |
| Llama3-8b-inst | 72963 | 0.92±0.03 | 0.94±0.02 | / | / |
| Llama3-8b-inst | 72972 | 0.92±0.03 | 0.94±0.02 | / | / |
| Llama3-8b-inst | 73017 | 0.91±0.03 | 0.94±0.02 | / | / |
| Llama3-8b-inst | 73042 | 0.92±0.03 | 0.94±0.02 | / | / |
| Llama2-70b-inst | 69087 | 0.74±0.12 | 0.90±0.08 | / | / |
| Llama2-70b-inst | 72578 | 0.75±0.12 | 0.90±0.09 | / | / |
| Llama2-70b-inst | 72600 | 0.71±0.13 | 0.91±0.09 | / | / |
| Llama2-70b-inst | 72869 | 0.71±0.13 | 0.91±0.09 | / | / |
| Llama2-70b-inst | 72958 | 0.71±0.13 | 0.90±0.09 | / | / |
| Llama2-70b-inst | 72963 | 0.72±0.13 | 0.89±0.10 | / | / |
| Llama2-70b-inst | 72972 | 0.70±0.13 | 0.88±0.10 | / | / |
| Llama2-70b-inst | 73017 | 0.70±0.13 | 0.87±0.10 | / | / |
| Llama2-70b-inst | 73042 | 0.71±0.13 | 0.88±0.10 | / | / |
| Llama2-7b-inst | 69087 | 0.56±0.05 | 0.56±0.05 | / | / |
| Llama2-7b-inst | 72578 | 0.57±0.05 | 0.55±0.05 | / | / |
| Llama2-7b-inst | 72600 | 0.57±0.05 | 0.56±0.04 | / | / |
| Llama2-7b-inst | 72869 | 0.57±0.05 | 0.56±0.04 | / | / |
| Llama2-7b-inst | 72958 | 0.57±0.05 | 0.56±0.04 | / | / |
| Llama2-7b-inst | 72963 | 0.57±0.05 | 0.57±0.05 | / | / |
| Llama2-7b-inst | 72972 | 0.57±0.05 | 0.56±0.05 | / | / |
| Llama2-7b-inst | 73017 | 0.57±0.05 | 0.56±0.05 | / | / |
| Llama2-7b-inst | 73042 | 0.57±0.05 | 0.56±0.05 | / | / |
| Llama3-70b-inst | 69087 | 0.90±0.08 | 0.92±0.09 | / | / |
| Llama3-70b-inst | 72578 | 0.92±0.08 | 0.92±0.09 | / | / |
| Llama3-70b-inst | 72600 | 0.92±0.08 | 0.93±0.09 | / | / |
| Llama3-70b-inst | 72869 | 0.93±0.07 | 0.93±0.08 | / | / |
| Llama3-70b-inst | 72958 | 0.93±0.07 | 0.94±0.08 | / | / |
| Llama3-70b-inst | 72963 | 0.92±0.08 | 0.93±0.09 | / | / |
| Llama3-70b-inst | 72972 | 0.91±0.08 | 0.93±0.09 | / | / |
| Llama3-70b-inst | 73017 | 0.92±0.08 | 0.94±0.09 | / | / |
| Llama3-70b-inst | 73042 | 0.91±0.09 | 0.94±0.08 | / | / |

## D.3 RESULTS PER BOOK

In Fig. 6, we provide the baseline results without text-specific memory separately for each of the 9 books in Book-SORT.

In Fig. 7, we provide the in-context memory results separately for each of the 9 books in Book-SORT.

## D.4 RELATIONSHIP BETWEEN IN-CONTEXT MEMORY RESULTS AND DISTANCE BETWEEN SEGMENTS ACROSS EXCERPT LENGTHS

In Fig. 8 and Fig 9, we show the average accuracy by the distance between segments for all the excerpt lengths and segment lengths.

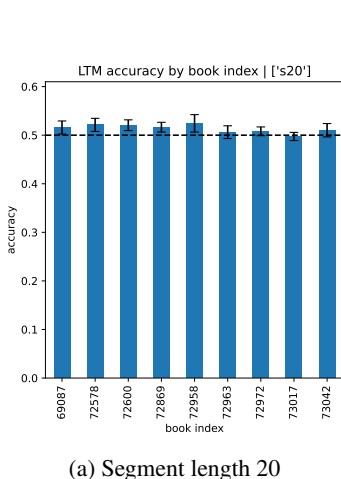
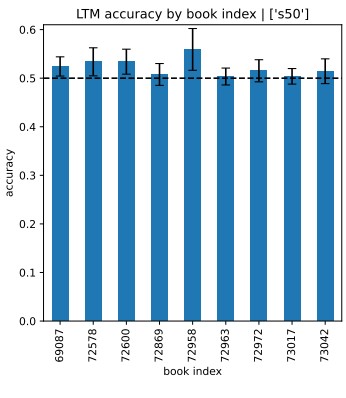

(a) Segment length 20                    (b) Segment length 50

Figure 6: Models' baseline performance by book (error bars indicate standard deviation)

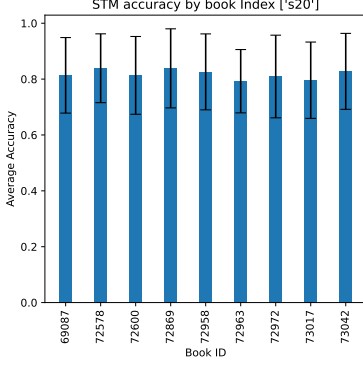
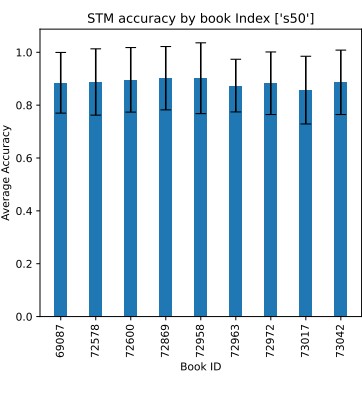

(a) Segment length 20                    (b) Segment length 50

Figure 7: Models' in-context memory performance by book (error bars indicate standard deviation)

## D.5 IN-CONTEXT MEMORY: FULL BOOK EVALUATION

Recent LLMs have longer context windows for which they can perform well. This opens the possibility to test in-context memory for these books when presented with a complete book in their context window. It is straightforward to create an adequate SORT-dataset to test how well an LLM can recall sequence order when presented with a full book instead of just an excerpt. Initial findings from evaluation of Llama3.1-8b-Instruct when presented with the complete book *The Murder of Roger Ackroyd* suggest that the model is performing worse than humans (see Figure 10). While the human performance shown as a reference in Figure 10 is based on reading the book up to 30 days prior to testing, the model only has the book-text in its context window without the presence of task-irrelevant additional episodes that occurred for the humans. To match the difficulty to human episodic memory testing, future versions of SORT datasets could introduce the addition of irrelevant documents, similar to a needle-in-a-haystack task (Kamradt, 2023).

The SORT-dataset for this evaluation is based on 300 pairs of 50-word segments for each of the segment-distance bins that we report. Because we test on a complete book and not an excerpt from a book, we modified the reading instruction of prompt 2 in Table 6 by replacing "excerpt from the book" with "the complete book".

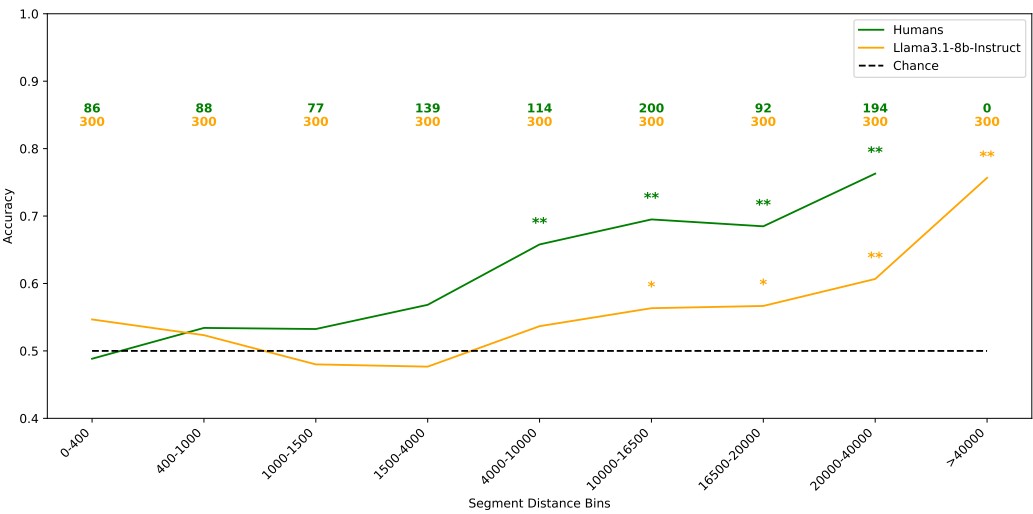

Figure 10: SORT evaluation of Llama3.1-8b-Instruct (128k context window) on a SORT dataset based on *The Murder of Roger Ackroyd*. Numbers indicate the sample-number within the bins. Asterisks indicate statistical significance: *p<0.05 and **p<0.01

## D.6 IN-CONTEXT MEMORY: LOST-IN-THE-MIDDLE EFFECT

The lost-in-the-middle effect is present in the in-context memory condition of SORT (Fig. 11). Both segments are considered to be in the middle section of the excerpt when the first word of each segment is in the middle one-third of the excerpt. Using logistic regression, we examined the interaction between whether both segments are in the middle section of the excerpt and the excerpt length, after controlling for the distance between the two segments. For nine out of ten models, we found a significant lost-in-the-middle effect, where accuracy is lower when both segments are in the middle section. For seven out of ten models, there is also a significant interaction between whether both segments are in the middle section of the excerpt and the excerpt length, suggesting that the degree of the lost-in-the-middle effect varies across excerpt lengths.

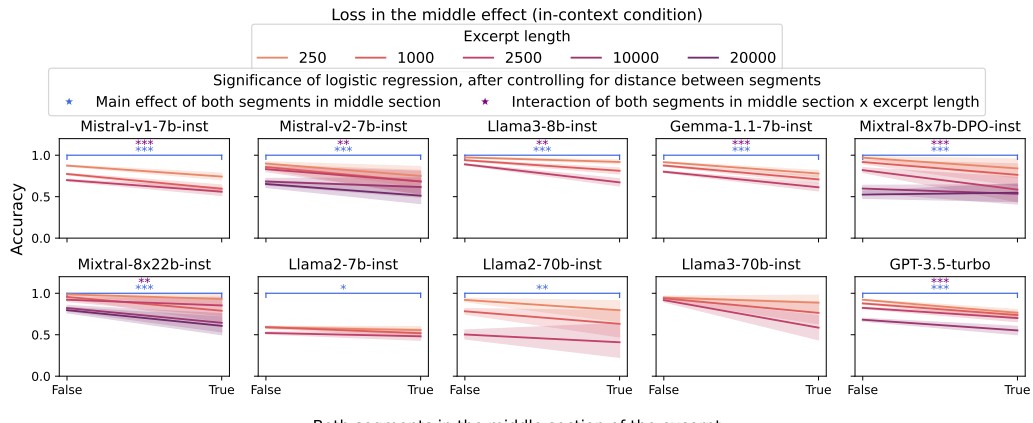

Figure 11: Lost-in-the-middle effect for the in-context memory condition

### D.7 IN-CONTEXT MEMORY: CONTEXT-EXTENDED LLAMA3-8B-INSTRUCT MODEL

We evaluate a single context-extended Llama3-8b-Instruct model (Zhang et al., 2024a) on SORT that reports high performance on other benchmarks after context-extension to see how performance changes. Our findings (Figure 12) show a strong degradation in performance with low performance on 20k word excerpts, despite a claimed support for 80k context windows and high performance on needle-in-a-haystack tests for long sequences. This highlights the need for high quality data during context-extension, and shows an additional use for SORT: to benchmark context-extension methods.

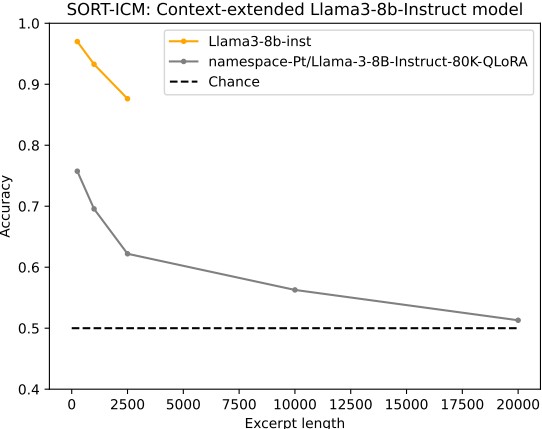

Figure 12: Evaluation of a context-extended model with a supposed 80k context support shows degraded performance.

### D.8 BASELINE PERFORMANCE

In Fig. 13, we provide the SORT results based on parametric memory for all models across various segment distances. Due to the recent addition of the texts in Book-SORT to the public domain, we expect that models were not trained on these texts, i.e. they should not have text-specific memory. Performance is higher for segment pairs that have a short distance and a high distance in the books, indicating that these are more likely to be sort-able without episodic memory, based on temporal order reasoning.

## E  BOOK-SORT RESULTS FROM ADDITIONAL MODELS

### E.1  BASE MODELS

We chose 2 base models to evaluate, Llama3-8b and Mistral-7b, whose fine-tuned versions (Llama3-8b-inst and Mistral-v2-7b-inst) performed well on SORT based on in-context memory. Figure 14 shows that both the base models got around chance performance across all the excerpt lengths and segment lengths.

### E.2  STATE-SPACE MODELS

We tested an instruction-tuned version of the state space model RWKV (Peng et al., 2023), available in Huggingface as RWKV/rwkv-raven-7b. The results of the prompt sweep on SORT with in-context memory yielded a performance of 51% – very close to chance levels. A possibility for this is a larger sensitivity to prompting, e.g. this model might require instructions to be given in a different order. We assume that this is due to insufficient instruction tuning. While it could be interesting to see the performance of a state-space model with memory other than in-context, we leave this question to future work.

## F  FINETUNING OF LLAMA3-8B-INSTRUCT

**Fine-tuning details.**    We fine-tuned Llama3-8b-Instruct and Mistral-7b-v0.2-Instruct on a single node with 8 A100 GPUs. The books (without pre-processing beyond removing Project Gutenberg related text, i.e. including chapter signifiers) are split into chunks of 5000 words and contextualized in the same way in which excerpts are presented in-context in our experiments, i.e. together with the book-title in a user prompt along with a preceding system prompt. For the instruction data, we exclude the following task types: "experience", "stylized_response", "joke", "trivia", "roleplay", "riddle" and "greeting". Samples containing both book-chunks and instruction-following examples are padded to the maximum length in a batch. The effective batch size in our experiments is 192. We choose a moderately low initial learning rate of 5e-6 with cosine decay and a small amount of weight decay set to 1e-4. The chunks of books comprise a total of 116 independent samples. Together with 3 500 instruction samples from the OpenHermes dataset (Teknium, 2023), this means 19 steps of gradient descent are taken in one epoch. We fine-tuned both models for a total of 5 epochs, however only the first epoch would qualify for episodic memory testing, since one of the characteristics of episodic memory is that it is single-shot learned (see definition 1 and Appendix A for a discussion). This precludes complete memorization of texts, for which multiple repetitions are needed (Ovadia et al., 2024).

**Inclusion of instruction data to avoid catastrophic forgetting.**    Fine-tuning an instruction-tuned model on specific data can lead to catastrophic forgetting (Luo et al., 2024), such that only a few steps of gradient descent can be enough to undo previous behavioral alignment (Qi et al., 2023; Zhan et al., 2024). To retain the general ability to follow instructions, and to allow for control condition fine-tuned models in which the book text is not part of the training data, we include 3, 500 instruction samples from the OpenHermes2.5 dataset on Huggingface (Teknium, 2023). Therefore the baseline without text-specific memory to compare with is not only the respective initial model before fine-tuning, but the same model when fine-tuned on the same 3, 500 instruction samples but excluding the 116 samples of book chunks.

**Overfitting does not lead to better performance on SORT.**    To test whether overfitting on the texts would lead to better performance, we finetuned a Llama3-8b-Instruct model for 120 and 300 repetitions of the same chunks of book text (mixed with 300k and 30k instruction samples respectively). We found that training with many repetitions of the book text, resulting in a sharper decrease in perplexity, also did not support the ability to recall the order of segments. This still holds when a SORT-task readout layer is trained on 8 out of 9 books and then evaluated on the held-out book (see Figure 15). Even though the perplexity for *The Murder of Roger Ackroyd* dropped from 9.9 to 2.4, the performance for SORT did not increase.

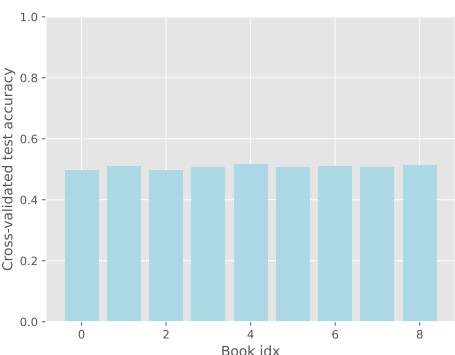

Figure 15: Crossvalidated task-readout performance for Llama3-8b-Instruct after finetuning on 120 repetitions of the book chunks, along with 300k instruction samples.

### F.1 PERPLEXITY ANALYSIS OF FINE-TUNED MODELS

To confirm that fine-tuning on the books makes a model learn about the segments, we compare the perplexities of the two segments shown in SORT without source text presented in-context. We find that when the models are finetuned on data that includes the chunks of the books, they have a substantially lower perplexity for both segments, compared with the models fine-tuned only on the instruction data (see figure 16). Note that the scale of these perplexity values highlights that our task is likely out of distribution, presumably with little to no similar instruction data seen during pre-training and fine-tuning.

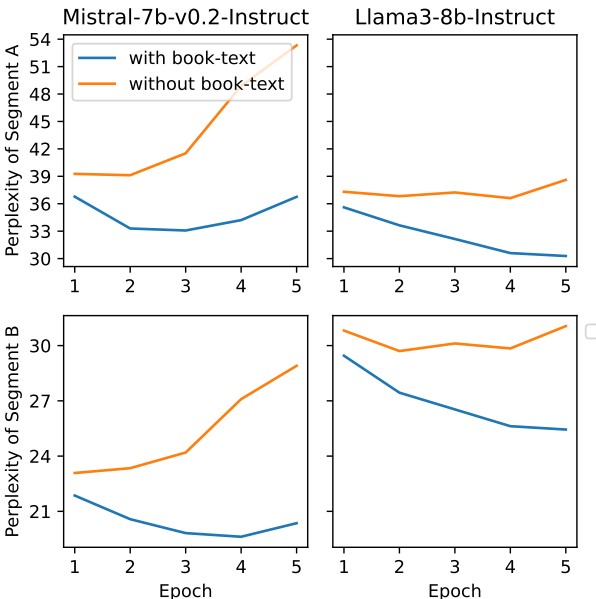

Figure 16: Perplexity of the two segments after fine-tuning of Mistral-7b-v0.2-Instruct and Llama3-8b-Instruct, when presented in the absence of in-context access to source excerpts.

### F.2 COMPARISON OF SORT PERFORMANCE AFTER FINE-TUNING USING MCNEMAR'S TEST

We find that even though the book-text finetuned Llama3-8b model has a form of memory of the books' texts, the epoch-matched performance between the models fine-tuned without the book-chunks does not differ statistically for any epoch (Figure 17). For this analysis we use McNemar's test

since we have an exact match of presented samples for both the memory-finetuned model and the baseline that does not form any memory of the text (Figure 16). We find high p-values, indicating no difference in performance between models fine-tuned with and without the book text (Figure 18), neither for Llama3-8b-Instruct, nor for Mistral-7b-v0.2-Instruct. Note that only epoch 1 qualifies to be tested for episodic memory since the books are only seen once (requirement 2 in Definition 1).

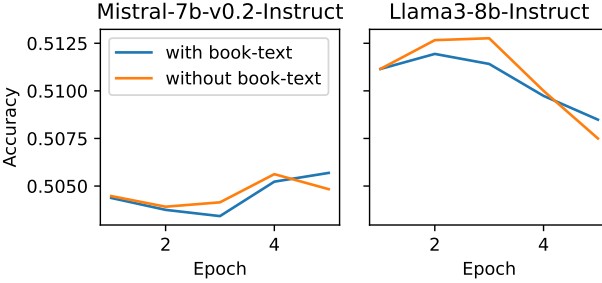

Figure 17: Accuracy of Llama3-8b-Instruct and Mistral-7b-v0.2-Instruct across epochs of finetuning on data including and excluding relevant book-text. Figure 18 shows that differences between accuracies shown here are not statistically significant (p>0.05).

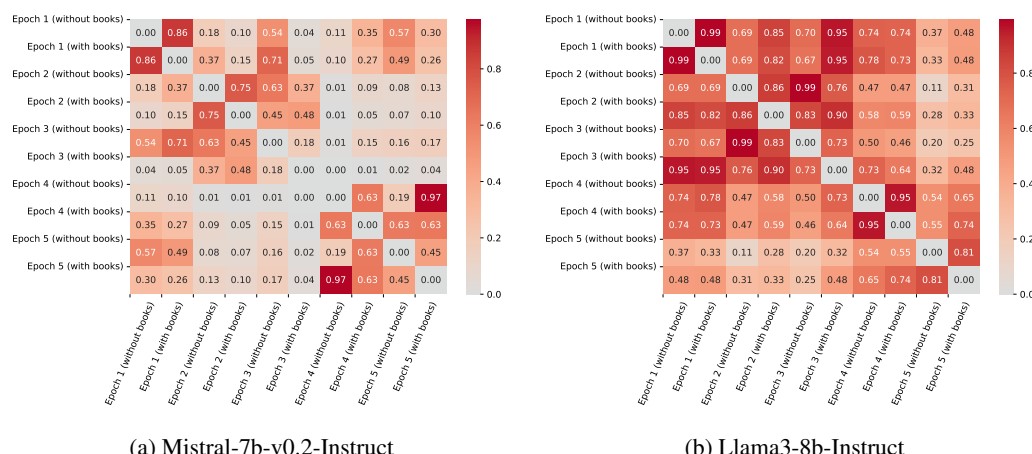

(a) Mistral-7b-v0.2-Instruct            (b) Llama3-8b-Instruct

Figure 18: McNemar's Test matrix of fine-tuned models performance. Shown are p-values indicating whether a model checkpoint (row) is different in its accuracy compared to another checkpoint (columns) with statistical significance. We fine-tuned with and without the books used in Book-SORT. There is no statistically significant difference between the models finetuned without and with book text. The effect of fine-tuning seems insignificant even without correcting these p-values for multiple comparisons.

### F.3 COMPARISON OF SORT PERFORMANCE AFTER FINE-TUNING USING A PAIRWISE T-TEST

Testing the binary correctness evaluated based on a greedily sampled token does not allow us to draw conclusions about sub-threshold effects of fine-tuning on task performance. To test whether the models fine-tuned on the books is better than the models that are fine-tuned without chunks from the books, we performed a pairwise t-test on a continuous measure of accuracy based on the token log-probabilities. We compute the likelihood of the correct answer by taking the log ratio of the correct answer among all answers that can be mapped to either A or B, i.e. we are interested in $\log\left(\frac{p(a=y)}{p(a=A)+p(a=B)}\right)$, where $y$ is the correct answer.

The results shown in figure 19 suggest that fine-tuned models do improve over the base model, with the book text condition performing better than the others after one epoch of training with statistical

significance ($p < 0.01$). Even though there is an effect, the magnitude is very small, as can be seen in Figure 20, and this positive effect could also be attributed to interleaving the instruction data with samples including longer texts ($5,000$ words) compared to just the instruction samples. Decreasing log-probability of the correct answer reflects catastrophic forgetting associated with training for multiple epochs on a small dataset, which violates requirement (2) of episodic memory (Definition 1). Figure 19 (second row for both matrices) shows that log-probabilities after the first epoch is statistically significantly higher for the fine-tuned models that included the book-text compared to all other epochs with and without including the book-text in fine-tuning. However, Figure 20 shows that the difference in log-probabilities of the correct answer is small after the first epoch, indicating that inserted memory does not affect explicit question answering ability about temporal order memory (see Figure 17), which is one of the requirements for episodic memory (req. 5 in definition 1).

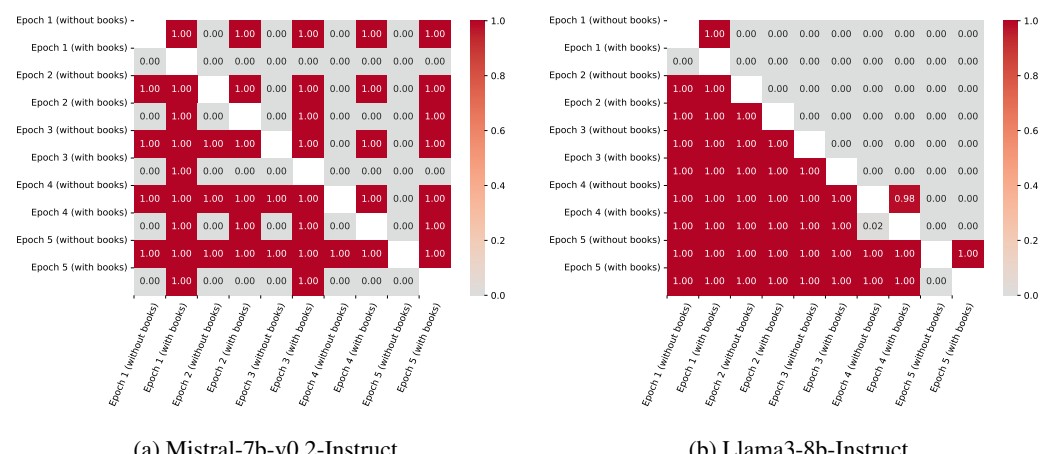

(a) Mistral-7b-v0.2-Instruct          (b) Llama3-8b-Instruct

Figure 19: Pairwise t-test matrix of fine-tuned models. Shown are p-values indicating whether a model (row) has higher log probabilities of the correct answer compared to another model (columns) with statistical significance. Row 2 is significantly better than all other epochs with or without book-text.

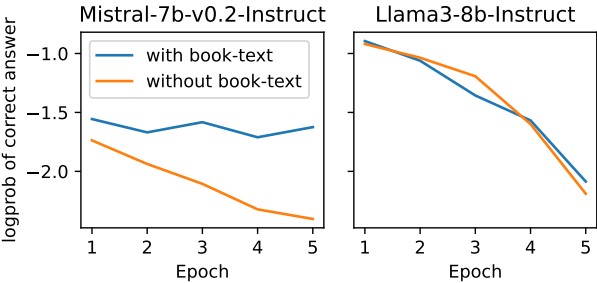

Figure 20: Log-probability of the correct answer for fine-tuned models across epochs. Figure F.3 shows statistical significance between conditions and epochs for this data.

## F.4 TRAINING ON SORT?

Our findings that fine-tuning with a language-modeling objective does not lead to temporal order memory might be due to the out-of-distribution property of the task - recalling the order of text is something that is not well covered in the models' training data distribution. Rather than implying that SORT is not suitable to test this type of memory, this highlights a lacking ability in current specific models to use parametric memory for this aspect of episodic memory: if they are asked about the order of a document that they have been trained on, they can in fact not recall it. Directly training a

model on parametric sequence order recall tasks might improve its performance, however we would not expect this to generalize to other aspects of episodic memory (Appendix A). In this work, we only evaluated existing models' capabilities (i.e. we did not modify models to become better at the task), and we suggest that for SORT to be maximally informative about general episodic memory capabilities of a model, it should only be used as a test-set task (not to be included in training and not to be optimized for specifically in terms of architecture and hyperparameters). Training on SORT risks that a less generalizable mechanism is used to specifically infer temporal order information, without generalizing to other types of relations between parts of a sequence (see Appendix A). This mirrors how we can explicitly include order information in RAG, which could also easily "solve" SORT, but would not be informative about RAGs suitability to function as an episodic memory insertion method for a model more broadly (see our discussion on RAG in section 6).

### F.5 IN-CONTEXT MEMORY PERFORMANCE OF FINE-TUNED MODELS

Despite the inclusion of instruction data in fine-tuning, the accuracy with source excerpts presented in-context of SORT decreased from 0.93 to 0.90 after a single epoch and to 0.88 after three epochs of fine-tuning for Llama3-8b-Instruct. For the instruction-data only baseline of Llama3-8b-Instruct, the performance degraded slightly less with an accuracy of 0.91 after the first epoch of fine-tuning.

## G  CODE AND DATA

Upon publication we will provide the code to create SORT datasets and evaluate models on SORT in a public GitHub repository, along with the Book-SORT dataset used in this work. Our evaluation code currently supports the OpenAI API, Huggingface Transformers (Wolf et al., 2020) and vLLM (Kwon et al., 2023) for distributed inference.

Experiment data from our Book-SORT evaluation is located in a Google Drive folder, along with the human experiment data. These will be made accessible openly through Huggingface datasets.

**License.**  We make our code and data openly available under a permissive BSD-3 license for code. Data including Book-SORT is available under a CC0 license.

Table 12: Accuracy and Difference of Various Models on Multiple Books at Excerpt Lengths of 20 and 50, with in-context memory (Part 2)

| Model name | Book | SORT S20 | SORT | SORT-Extend S20 | SORT-Extend S50 |
|---|---|---|---|---|---|
| Mixtral-8x7b-DPO-inst | 69087 | 0.86±0.10 | 0.87±0.13 | 0.63±0.18 | 0.49±0.14 |
| Mixtral-8x7b-DPO-inst | 72578 | 0.88±0.10 | 0.90±0.10 | 0.63±0.18 | 0.57±0.14 |
| Mixtral-8x7b-DPO-inst | 72600 | 0.89±0.10 | 0.91±0.10 | 0.63±0.18 | 0.58±0.15 |
| Mixtral-8x7b-DPO-inst | 72869 | 0.90±0.09 | 0.92±0.10 | 0.61±0.17 | 0.55±0.15 |
| Mixtral-8x7b-DPO-inst | 72958 | 0.90±0.09 | 0.93±0.09 | 0.57±0.16 | 0.57±0.15 |
| Mixtral-8x7b-DPO-inst | 72963 | 0.89±0.10 | 0.92±0.10 | 0.56±0.16 | 0.55±0.15 |
| Mixtral-8x7b-DPO-inst | 72972 | 0.89±0.10 | 0.91±0.09 | 0.55±0.16 | 0.54±0.15 |
| Mixtral-8x7b-DPO-inst | 73017 | 0.87±0.10 | 0.91±0.10 | 0.55±0.14 | 0.54±0.14 |
| Mixtral-8x7b-DPO-inst | 73042 | 0.87±0.10 | 0.91±0.09 | 0.57±0.14 | 0.55±0.14 |
| Mixtral-8x22b-inst | 69087 | 0.92±0.08 | 0.93±0.09 | 0.73±0.13 | 0.73±0.11 |
| Mixtral-8x22b-inst | 72578 | 0.92±0.08 | 0.95±0.07 | 0.76±0.12 | 0.76±0.12 |
| Mixtral-8x22b-inst | 72600 | 0.93±0.08 | 0.96±0.07 | 0.77±0.12 | 0.78±0.11 |
| Mixtral-8x22b-inst | 72869 | 0.93±0.08 | 0.97±0.07 | 0.78±0.12 | 0.80±0.11 |
| Mixtral-8x22b-inst | 72958 | 0.93±0.08 | 0.97±0.06 | 0.79±0.12 | 0.80±0.11 |
| Mixtral-8x22b-inst | 72963 | 0.92±0.09 | 0.97±0.06 | 0.78±0.12 | 0.78±0.12 |
| Mixtral-8x22b-inst | 72972 | 0.92±0.09 | 0.97±0.07 | 0.78±0.12 | 0.79±0.12 |
| Mixtral-8x22b-inst | 73017 | 0.93±0.09 | 0.97±0.07 | 0.78±0.12 | 0.79±0.12 |
| Mixtral-8x22b-inst | 73042 | 0.93±0.09 | 0.97±0.07 | 0.78±0.12 | 0.79±0.12 |
| Mistral-v2-7b-inst | 69087 | 0.85±0.10 | 0.87±0.11 | 0.64±0.15 | 0.66±0.13 |
| Mistral-v2-7b-inst | 72578 | 0.85±0.11 | 0.87±0.10 | 0.63±0.15 | 0.65±0.14 |
| Mistral-v2-7b-inst | 72600 | 0.86±0.11 | 0.87±0.10 | 0.64±0.14 | 0.67±0.14 |
| Mistral-v2-7b-inst | 72869 | 0.85±0.11 | 0.87±0.11 | 0.64±0.15 | 0.68±0.13 |
| Mistral-v2-7b-inst | 72958 | 0.86±0.10 | 0.88±0.11 | 0.65±0.15 | 0.68±0.14 |
| Mistral-v2-7b-inst | 72963 | 0.83±0.11 | 0.88±0.11 | 0.64±0.14 | 0.68±0.14 |
| Mistral-v2-7b-inst | 72972 | 0.84±0.11 | 0.88±0.10 | 0.63±0.14 | 0.68±0.14 |
| Mistral-v2-7b-inst | 73017 | 0.83±0.11 | 0.88±0.10 | 0.63±0.14 | 0.68±0.14 |
| Mistral-v2-7b-inst | 73042 | 0.83±0.11 | 0.88±0.10 | 0.63±0.14 | 0.68±0.14 |
| Mistral-v1-7b-inst | 69087 | 0.74±0.04 | 0.82±0.03 | / | / |
| Mistral-v1-7b-inst | 72578 | 0.75±0.04 | 0.81±0.03 | / | / |
| Mistral-v1-7b-inst | 72600 | 0.74±0.04 | 0.80±0.03 | / | / |
| Mistral-v1-7b-inst | 72869 | 0.74±0.04 | 0.81±0.03 | / | / |
| Mistral-v1-7b-inst | 72958 | 0.74±0.04 | 0.81±0.03 | / | / |
| Mistral-v1-7b-inst | 72963 | 0.74±0.04 | 0.80±0.03 | / | / |
| Mistral-v1-7b-inst | 72972 | 0.75±0.04 | 0.80±0.03 | / | / |
| Mistral-v1-7b-inst | 73017 | 0.74±0.04 | 0.80±0.03 | / | / |
| Mistral-v1-7b-inst | 73042 | 0.75±0.04 | 0.80±0.03 | / | / |
| Gemma-1.1-7b-inst | 69087 | 0.82±0.03 | 0.88±0.03 | / | / |
| Gemma-1.1-7b-inst | 72578 | 0.83±0.04 | 0.89±0.03 | / | / |
| Gemma-1.1-7b-inst | 72600 | 0.83±0.04 | 0.88±0.03 | / | / |
| Gemma-1.1-7b-inst | 72869 | 0.84±0.04 | 0.89±0.03 | / | / |
| Gemma-1.1-7b-inst | 72958 | 0.84±0.04 | 0.89±0.03 | / | / |
| Gemma-1.1-7b-inst | 72963 | 0.84±0.04 | 0.88±0.03 | / | / |
| Gemma-1.1-7b-inst | 72972 | 0.84±0.04 | 0.87±0.03 | / | / |
| Gemma-1.1-7b-inst | 73017 | 0.83±0.04 | 0.87±0.03 | / | / |
| Gemma-1.1-7b-inst | 73042 | 0.84±0.04 | 0.87±0.03 | / | / |
| GPT-3.5-turbo | 69087 | 0.86±0.03 | 0.88±0.03 | / | 0.69±0.04 |
| GPT-3.5-turbo | 72578 | 0.87±0.03 | 0.89±0.03 | / | 0.69±0.04 |
| GPT-3.5-turbo | 72600 | 0.87±0.03 | 0.89±0.03 | / | 0.67±0.04 |
| GPT-3.5-turbo | 72869 | 0.87±0.03 | 0.90±0.03 | / | 0.67±0.04 |
| GPT-3.5-turbo | 72958 | 0.87±0.03 | 0.90±0.03 | / | 0.67±0.04 |
| GPT-3.5-turbo | 72963 | 0.86±0.03 | 0.89±0.03 | / | 0.67±0.04 |
| GPT-3.5-turbo | 72972 | 0.86±0.03 | 0.88±0.03 | / | 0.67±0.04 |
| GPT-3.5-turbo | 73017 | 0.85±0.03 | 0.88±0.03 | / | 0.67±0.04 |
| GPT-3.5-turbo | 73042 | 0.85±0.03 | 0.88±0.03 | / | 0.67±0.04 |

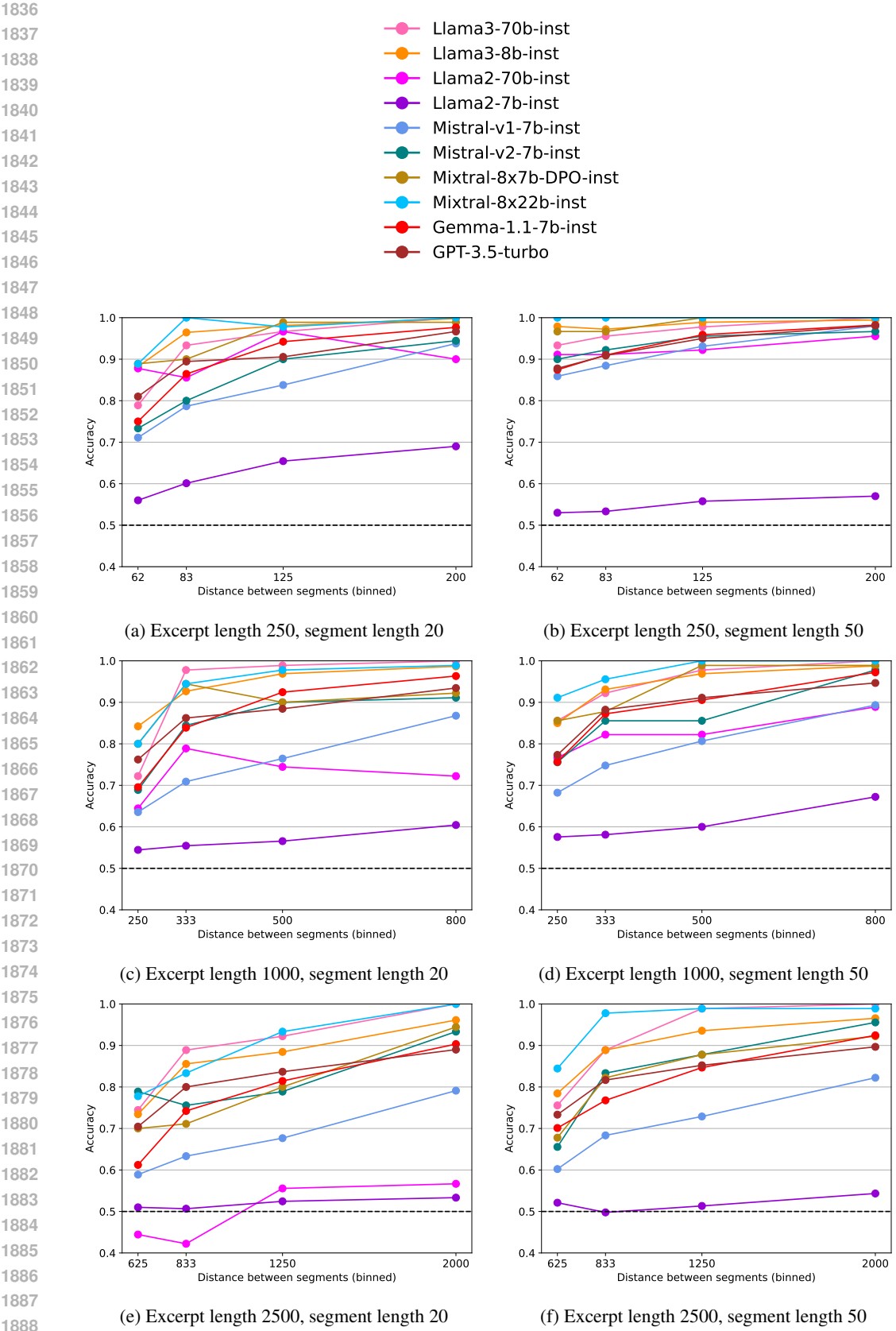

Figure 8: Average accuracy by distance between segments (All excerpt length), part A.

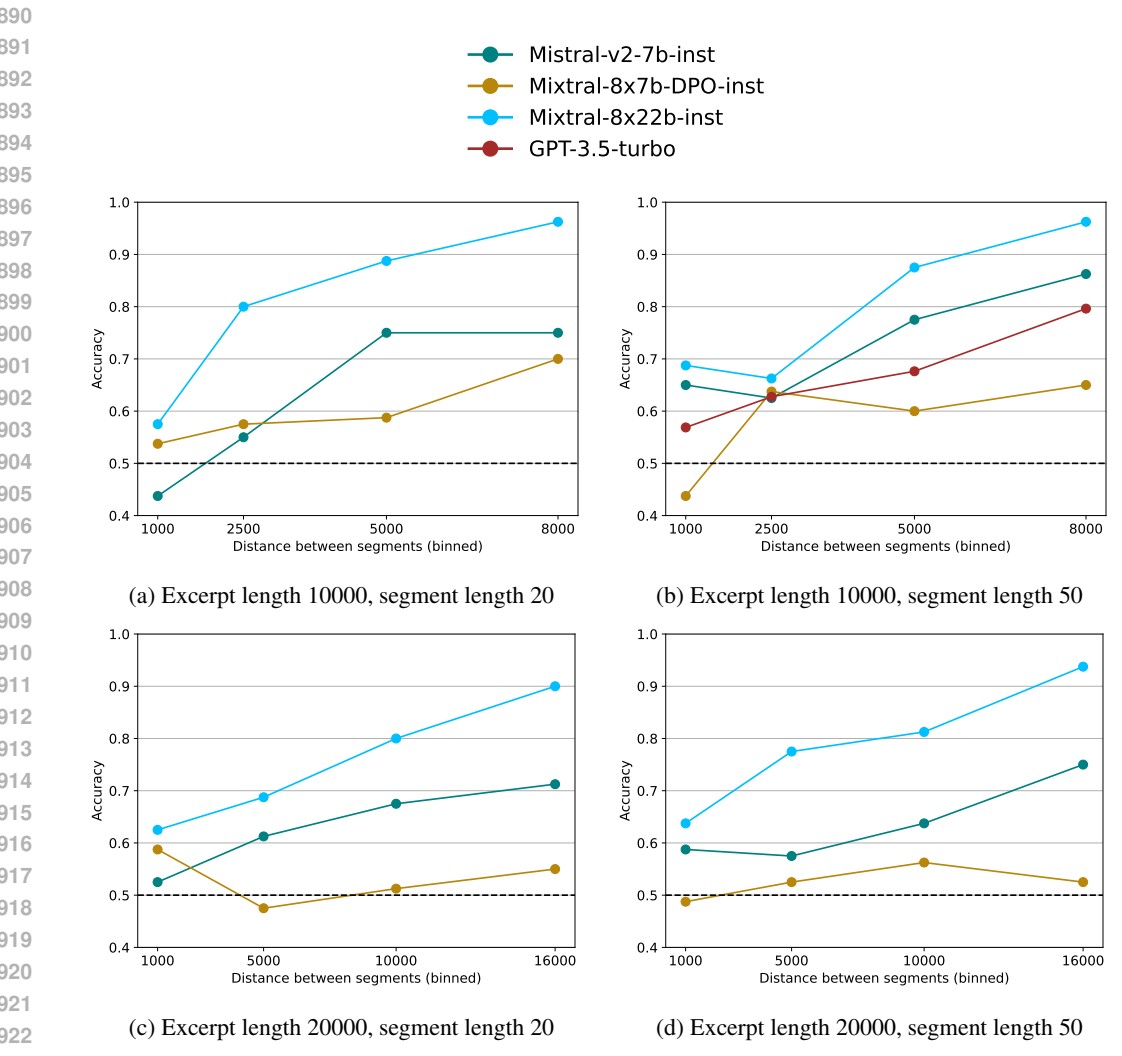

(a) Excerpt length 10000, segment length 20

(b) Excerpt length 10000, segment length 50

(c) Excerpt length 20000, segment length 20

(d) Excerpt length 20000, segment length 50

Figure 9: Average accuracy by distance between segments (All excerpt length), part B.

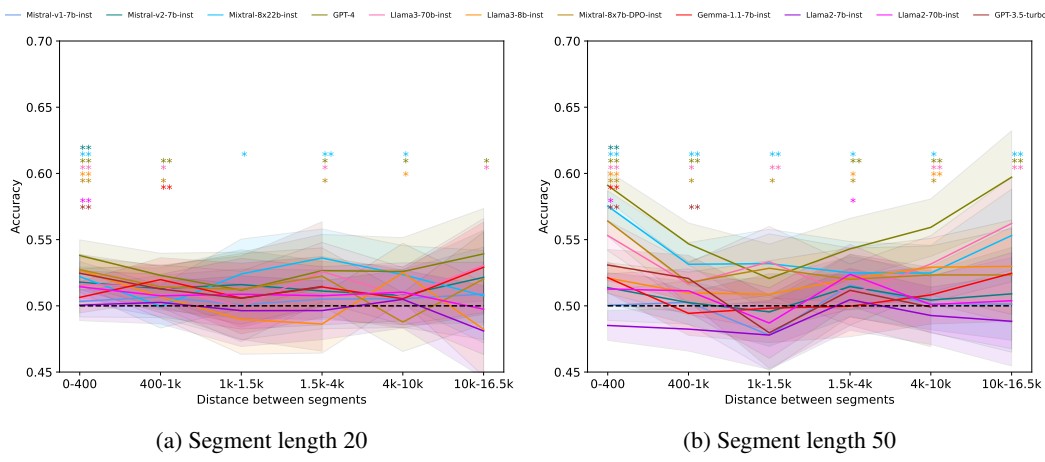

(a) Segment length 20

(b) Segment length 50

Figure 13: Baseline model performance on SORT without text-specific memory by segment distance (95% bootstrapped confidence interval). Significant difference from chance is marked with asterisks (*p-value<0.05,**p-value<0.01).

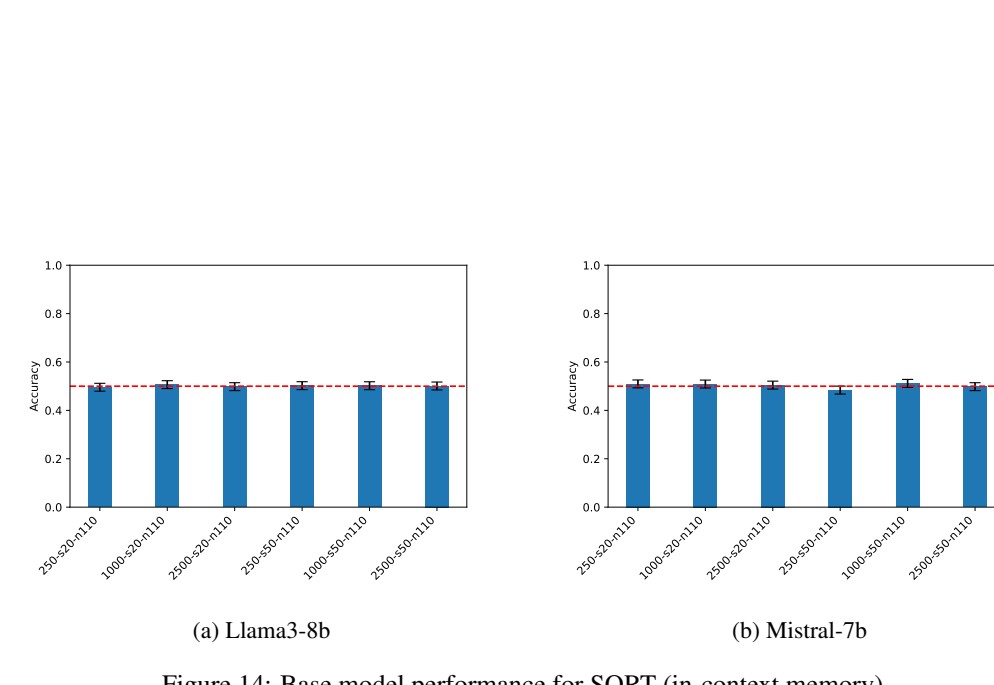

(a) Llama3-8b             (b) Mistral-7b

Figure 14: Base model performance for SORT (in-context memory).

