# OpenReview forum: "Assessing Episodic Memory in LLMs with Sequence Order Recall Tasks"
_ICLR.cc/2025/Conference — Submitted to ICLR 2025_

### Official Review · Reviewer_7fun · 2024-10-27

**Soundness:** 3
**Presentation:** 3
**Contribution:** 3
**Rating:** 8
**Confidence:** 4

**Summary:**

The paper proposes a new benchmark, SORT, for testing the episodic memory of and temporal order reasoning of LLMs, which is lacking in the current evaluation benchmarks. This test is inspired by cognitive sciences, and composed of pairs of randomly selected non-overlapping passages from fiction books, with a length of 20-50 words, which the model is provided with, along with contextual information in the form of an excerpt from the book, or without it. It is then asked to decide which of them precedes the other. Several setups are tested: Zero-shot prompting with and without context, with RAG, and finetuned on book data as a way to insert memory. The performance of the several models is compared to that of human baselines on this task (evaluated using 155 participants).
The results show that without context models fail entirely in this task (with performance in the range of 50% - random chance). When given context, models succeed very well, but their performance degrades as the length of the excerpt increases. Finetuning on language modeling task with the book data does not improve the results much.
Due to these results, specifically degradation of performance with context length, authors surmise that the transformer’s context is more similar to working memory than episodic memory. Naive RAG can also not be considered episodic memory since it does not preserve order, and indeed performance is not high with this implementation. Order preserving RAG, with an oracle retriever, is successful.

**Strengths:**

* The authors builds a new task for assessing episodic memory which is a novel test for LLMs. This is done in a scalable way which can be used to generate more information if needed, and is furthermore carried out in a way that minimizes contamination.
* By using a no-context baseline, the authors show that the benchmark cannot be solved from temporal reasoning alone without knowing the context (~50% accuracy).
* Human evaluations are used to provide a human baseline on this task, which is very effective in understanding the limitations and successes of LLMs.
* The authors compare the most common strategies used to adapt LLMs to specific tasks, and thus are able to compare and contrast them, which is also helpful when designing solutions based on LLMs that require this capability (specifically, pointing out that RAG naively does not perform well and that pretraining on book data does not help).
* The authors compare several smaller and larger models (mostly open source) on this task which shows that performing well on this task is not just a matter of model size.
* Authors perform prompt-optimization per model to reduce biases
Overall, many comparisons across different scenarios allow for a comprehensive understanding of LLM performance on this task.

**Weaknesses:**

* It is very clear that the in-context scenario is relatively simple for LLMs. The paper thus hypothesizes that this is more similar to working memory. It would be interesting to compare to baseline human performance when shown the in-context SORT task, as a way of strengthening the claim.
* Other than the fine tuning setup, the model cannot be said to have seen the entire book in any of the setups (this may be possible with long-context). This is markedly different from the human baseline, which also relies on long-term memory which makes comparison (and possibly relating the task to episodic memory) a bit more difficult.

**Questions:**

* Have you considered long-context models as a way of showing the model the entire book in context which may be more similar to the actual setup in humans?
* Why is GPT-4 present in the baseline tests but missing in RAG and In-context setups? It seems to have performed very well in the baseline.

---

> ### Author Response · Authors · 2024-11-22
> **Clarification of design choices and new in-context memory experiment on a complete book**
>
> We thank you for your appreciative words and would like to respond to your concerns regarding the comparison between in-context memory and human performance on the task.
>
> ### **Mismatch between human experiment and in-context memory setup**
> We agree that testing short-term memory in humans using excerpts from a book could yield valuable insights. However, our broader objective is to evaluate episodic memory in models, which we believe is better served by evaluating their memory of an entire book. This approach replicates the human scenario in our experiment, where participants had read the book up to 30 days prior, potentially with intervening irrelevant information.
>
> The current version of BookSORT uses smaller excerpts due to limitations in context length available during the initial experiments. At that time, open-source models with extended context windows were not yet accessible.
> To address this, **we are now conducting a new experiment using Llama3.1-8b-Instruct when given a complete book**. In this setup, we evaluate memory of *The Murder of Roger Ackroyd* (the same book used in our human experiment), using a new SORT dataset that encompasses the entire book. Results will be shared as soon as the experiment is completed.
>
>
> ### **Absence of GPT-4 results in memory-insertion experiments**
>
> While we included GPT-4 in our baseline evaluations to illustrate the need for memory of the books, we opted not to use closed-source models in our memory-insertion experiments. This decision reflects concerns about computational cost, lack of transparency regarding architecture and training processes, and potential modifications such as pruning or quantization.
> We believe closed-source models are best used to show what is currently possible with a state-of-the-art model. For our in-context memory experiments, open models already demonstrated strong results, and we attribute the low performance of vanilla RAG to its decontextualized nature (encoding and retrieving chunks independently), not to the LLM itself.
>
> We hope this clarifies our design choices! We will provide an update with the new results for Llama3.1-8b-Instruct when evaluated on the entire book as soon as the experiments are complete.

---

> > ### Author Response · Authors · 2024-11-26
> > **New full book in-context memory results**
> >
> > ### Results for Llama3.1-8b-Instruct (128k window) on The Murder of Roger Ackroyd
> >
> > We evaluated Llama3.1-8b-Instruct on the complete book that humans had read less than 30 days prior to our experiment. The results (which we now describe in Appendix C.5) show that performance is at chance level for segment-distances up to 10,000 and is lower than human performance. This is the case even though humans had plenty of task-irrelevant experience in between finishing the book and participating in our experiment. Future SORT datasets could include the presentation of multiple documents (e.g. multiple books concatenated or interleaved).

---

### Official Review · Reviewer_yhUH · 2024-10-30

**Soundness:** 3
**Presentation:** 3
**Contribution:** 3
**Rating:** 6
**Confidence:** 4

**Summary:**

The paper introduces a new evaluation framework, Sequence Order Recall Tasks (SORT), designed to assess episodic memory in large language models (LLMs). Unlike traditional benchmarks that focus on semantic memory, SORT requires models to recall the order of text segments, assessing memory that links content to its context. The authors created Book-SORT, a dataset of 36,000 text segment pairs from nine public domain books, and conducted human experiments, showing that humans can remember sequence order from long-term memory with notable accuracy. Their findings indicate that while models perform well with in-context memory, they struggle with SORT when only trained on book content, highlighting current LLMs' limitations in true episodic memory.

**Strengths:**

1.  The paper offers a simple yet effective task, easily understandable and extensible to various settings. The general formulation of SORT enables adaptation to multiple domains and sequential data types, such as video or audio, enhancing clarity and future applicability.
2. The authors have open-sourced both the Book-SORT dataset and the code, ensuring reproducibility and enabling further research by the community.
3. The paper features multiple experiment setting, covering in-context memory, Retrieval-Augmented Generation (RAG), and fine-tuning methods.

**Weaknesses:**

1. Although the SORT task is positioned as an assessment of episodic memory, the simple task of sequence order recall does not fully capture the complexities of episodic memory as seen in humans. Episodic memory typically involves more than the temporal order of events; it also includes contextual elements such as the significance, location, or source of memories. As currently designed, SORT assesses temporal ordering without requiring models to associate segments with rich context or situational details.
2. The authors note in the related work that sequence order prediction tasks have been used in prior language models, such as BART, where sequence reordering is part of the training objective. SORT may therefore be too easy to offer a strong evaluation of advanced memory capabilities in LLMs.
3. While the paper includes a broad range of experimental settings (in-context memory, RAG, fine-tuning), it does not clearly identify which specific factors most significantly improve performance on the SORT task. Table 2 indicates that with in-context settings, models can achieve near-perfect accuracy (e.g., Mixtral-8x22b-inst reaches 0.95 accuracy). This high performance suggests that in-context memory alone might be sufficient for solving SORT.

**Questions:**

1. Given the results in Table 2, it appears that larger context windows may positively impact model accuracy on the SORT task. However, many of the models used in this study are limited to an 8k context window. Could the authors consider incorporating models designed for extended context lengths, such as LongLLaMA[1], to assess whether increasing the context window further improves SORT accuracy?
2. In the RAG setup, how sensitive are the SORT results to the choice of embedding model and vector dimensionality?
3. When using in-context learning for the SORT task, does the model exhibit a "Lost in the Middle" effect, where it struggles to differentiate the order of segments that appear in the middle of longer contexts?

[1] Tworkowski S, Staniszewski K, Pacek M, et al. Focused transformer: Contrastive training for context scaling[J]. Advances in Neural Information Processing Systems, 2024, 36.

[2] Liu N F, Lin K, Hewitt J, et al. Lost in the middle: How language models use long contexts[J]. Transactions of the Association for Computational Linguistics, 2024, 12: 157-173.

---

> ### Author Response · Authors · 2024-11-22
>
> We are thankful for your thoughtful review and your suggestions. Along with some clarifications, we provide a new experiment and a new analysis of our in-context results to answer two of your questions.
>
> ### **New evaluation of a context-extended Llama3-8b-Instruct model (Appendix C.7)**
> We now report results for a context-extended model (Zhang et al., 2024a) and find that its performance on SORT is much worse than before context-length-extension on short excerpts, while only achieving 52% accuracy on excerpts with 20k words.
>
> ### **New analysis of lost-in-the-middle effect (Appendix C.6)**
> A new analysis (see Appendix C.6) shows that the lost-in-the-middle effect is present in the in-context memory evaluation with SORT. Using logistic regression, we examined the interaction between whether both segments are in the middle section of the excerpt and the excerpt length, while controlling for the distance between the two segments. For nine out of ten models, we found a significant lost-in-the-middle effect, where accuracy is lower when both segments are in the middle section. There is also a significant interaction between the excerpt length and whether segments are in the middle for seven out of ten models, suggesting that the degree of the lost-in-the-middle effect varies across excerpt lengths.
>
> ### **Embedding dimensionality in RAG**
>
> We do not expect a larger dimensionality in RAG to lead to better temporal order memory. This is because in vanilla RAG, segments are encoded independently of one another, thereby losing relative positional information that could even be encoded into the vectors. However, we expect that a larger dimensionality and a better chunk-encoding model would lead to a higher precision in retrieving chunks containing the segment text.
>
> ### **Defining Episodic Memory**
> To improve clarity, we have updated the manuscript to include a definition of episodic memory in LLMs. We think this will be useful to address your questions.
>
> > **Definition 1.1: Episodic Memory in LLMs**
> > Episodic memory refers to knowledge in a language model that:
> > 1) is specific and unique to a particular sequence;
> > 2) is acquired through a single exposure to that sequence (single-shot learning);
> > 3) contains information about the unique context of segments within that sequence (e.g.
> the temporal and general composition of events/items within the complete sequence);
> > 4) can still be retrieved when arbitrarily many tokens are processed in between the
> sequence’s encoding and its retrieval;
> > 5) has functional implications, meaning the knowledge can be used by the model to
> answer explicit queries.
>
> ### **Assessing temporal context in episodic memory with SORT**
>
> Requirement (3) in our definition describes a central characteristic of episodic memory, which requires it to not only store independent items/events/statements but instead requires memory of how they relate to their complete surrounding context. One such relation is the temporal order between segments, which we test in SORT. Even though this is just one aspect of episodic memory, it is the most studied aspect of episodic memory.
> SORT is an initial and easily extendable benchmark for episodic memory. In other existing datasets that involve reading comprehension, it is not clear how much relations between segments in the same context matter (often only very local and atomic relations matter). We agree that future work should focus on providing even more evaluation tasks that focus on testing different aspects of requirement (3) in our definition. This will help to assess more aspects of contextualized retrieval in LLM memory (e.g. of inferred location, relative significance etc.) and one direction for future work could be to filter existing datasets for samples in which requirement (3) is especially relevant.
>
> ### **For which LLM-architectures can SORT be used to evaluate episodic memory (in-context)?**
>
> We are aware that performance on SORT can be easily “hacked” by designing an architecture or specific prompting methods to be good at recalling the order of segments in a text. This means that SORT is not equally useful as an evaluation of episodic memory for all types of models and all training objectives and prompts. We intend SORT as an evaluation of causal/autoregressive LLMs, which excludes BART due to its bi-directional encoder, and have updated the text of the paper to reflect this.
>
> ### **Difficulty of SORT for in-context memory**
>
> The high performance in the in-context setting only holds for short excerpts (up to 2500 words). While SORT is not only meant to evaluate in-context memory, the SORT-ICM task can be made as difficult as desired by evaluating on longer texts, or merging multiple texts (see requirement 4 in our definition). We provide code that allows to easily create such SORT datasets.
>
> We thank you again for your comments and hope that we could answer your questions and clarify the main points you've raised!

---

> > ### Comment · Reviewer_yhUH · 2024-11-24
> > **Response to Author 12328**
> >
> > Thank you for your detailed and thoughtful response to my review. I appreciate the effort you've put into addressing the concerns I raised.

---

### Official Review · Reviewer_kx1o · 2024-11-03

**Soundness:** 2
**Presentation:** 1
**Contribution:** 1
**Rating:** 3
**Confidence:** 4

**Summary:**

The paper introduces a new evaluation task called the Sequence Order Recall Task (SORT) to assess episodic memory in large language models (LLMs). SORT requires models to recall the correct order of two text segments extracted from longer sequences. The authors create a dataset named Book-SORT, consisting of 36,000 pairs of text segments from nine books recently added to the public domain. They also conduct a human evaluation with 155 participants who had recently read one of the books, demonstrating that humans can recall some segment orders based on long-term memory, especially when the segments are far apart. The paper evaluates various LLMs on SORT using different methods of memory insertion, such as in-context learning, fine-tuning, and retrieval-augmented generation. The main finding is that models perform relatively well when given relevant context but fail to recall the order when using only their parametric memory.

**Strengths:**

- **Valuable Human Evaluation**: The paper conducts a large-scale human evaluation involving 155 participants who recently read a book. This provides valuable data on human long-term memory performance in recalling the order of text segments.

- **Highlighting Limitations of Parametric Memory**: Although simple, the paper rightfully brings attention to the limitations of current LLMs' parametric memory in handling one of the episodic memory tasks (here segment order). This is important for raising awareness and fostering further research in this area.

**Weaknesses:**

- **Little coverage of episodic memory** The correct order of text segments is one tiny aspect in a large spectrum of episodic memory capabilities: (i) places/spaces in which events occured, the (ii) intricate sorting of events and entities that are tracked along long periods of time. For example, we are able not only to sort the major events of our life, but also the major events that happened within the lifetime of a six-months long project, or the major events that happened in a movie, or a novel. We are able to sort, and recall the spatial and action details etc.


- **Reported events vs. "encountered" events** For humans, and LLMs in particular (for which language is also a *reporting tool* of episodic events), there are other aspects that matter: For example, what are the actual order of events as they happened in time (as reported by language), not only as they were told to the reader/LLM sequentially. Does it make sense to ask a model which book appeared first in its training set? but if so, based on which epoch? Are LLMs equipped to answer this type of questions? Wouldn't this task simply only test the positional encoding/embedding abilities of LLMs? because that's the only fundamental signal that allows them to know what comes before what.


- **Benchmark contamination** Although recently added to the public domain, the books are old and there are chances that they are part of the training data. I appreciate that the authors did the effort of verifying that the models failed to answer using only their parametric memory (e.g. Table1), but one cannot rule out that pre-training could have helped in the in-context task.


- **No significant technical innovation in methodology**

**Questions:**

- Older versions of GPT4 used to know by heart some of the books on the gutenberg project ("the Murder of Roger Ackroyd" is present there). Did you ensure that the models did not know the book? (e.g. by asking questions?)

- Why is the evaluation on GPT4-o missing from the second part? Would it be possible to include GPT-4 results or explain their omission?

- In which way your work is different from simply testing positional encoding capabilities of LLMs? Would it be possible to add analysis distinguishing positional encoding from true temporal understanding?

---

> ### Author Response · Authors · 2024-11-22
> **Why does SORT test non-abstractive sequence order recall?**
>
> We thank you for your comments!
>
> ### **What needs to be tested when assessing episodic memory**
> To improve clarity, we have updated the manuscript to include a first definition of episodic memory in LLMs. We hope this will also be useful to address your questions about our coverage of episodic memory.
>
> > **Definition 1.1: Episodic Memory in LLMs**
> > Episodic memory refers to knowledge in a language model that:
> > 1) is specific and unique to a particular sequence;
> > 2) is acquired through a single exposure to that sequence (single-shot learning);
> > 3) contains information about the unique context of segments within that sequence (e.g.
> the temporal and general composition of events/items within the complete sequence);
> > 4) can still be retrieved when arbitrarily many tokens are processed in between the
> sequence’s encoding and its retrieval;
> > 5) has functional implications, meaning the knowledge can be used by the model to
> answer explicit queries.
>
> ### **Coverage of episodic memory**
>
> Of course, temporal order is not the only type of relational context of segments that episodic memory covers.
>
> (i) Memory of spatial locations
>
> Spatial locations can be explicitly mentioned (e.g. book-text that contains the text “The school was located next to the river.”). In this case, recall would not be considered a good test for episodic memory, since it is very close to the retrieval of atomic facts (requirement 3 in our definition). A second type of spatial position information would only be given in agentic systems that need to navigate an environment. This would be an exciting setup and will become highly relevant, but it is not the focus of our current work, which looks at LLMs as they process documents.
>
> (ii) Memory of more abstract events
>
> SORT focuses on ordering of non-abstractive segments (segments of the exact text). We could construct a SORT task where the entities to be sorted are abstractive items (see requirement 3). Unlike in SORT, this involves a labor-intensive annotation process or a highly artificial sequence to be used (unlike more natural texts that can be used in SORT). We agree that these will be interesting extensions of SORT but we think that non-abstractive SORT has many advantages and will be useful for testing episodic memory.
>
> ### **Does SORT (for in-context memory) only test positional encoding?**
>
> > Does it make sense to ask a model which book appeared first in its training set? but if so, based on which epoch? Are LLMs equipped to answer this type of questions?
>
> Modern LLMs are usually trained for a single epoch, so texts are only encountered once unless duplicates are present (which is true for many texts). Due to the i.i.d. nature of training, we do not think it’d be possible for an LLM to (explicitly) know the order in which it has been trained on text. We therefore focus on intra-document temporal order memory to test episodic memory. Unless two books are presented in the same context window, it does not make sense to evaluate their order (in current architectures with current training methods).
>
> > **Reported events vs. "encountered" events**
>
> Episodic memory is indeed about the “encountered” temporal order, not the reported temporal order. Likewise, the in-context memory SORT task tests for memory of relative temporal position and retrieving and comparing past positional encoding associated with the two segments would present one way to achieve high performance. SORT can however be used to assess models that do not use a positional encoding and need to rely on other mechanisms (e.g. future Mamba models with strong instruction-following abilities).
>
> 1/2

---

> > ### Author Response · Authors · 2024-11-22
> > **Benchmark contamination, GPT-4 and technical innovation**
> >
> > ### **Benchmark contamination**
> >
> > This limitation of our dataset arises when models are evaluated of which we do not know their training data. To mitigate this issue, in internal tests, new copyrighted texts could be used as a way to verify the validity of results. Ideally, future work on the evaluation of memory capabilities in models will be centered around models for which the training data is known.
> >
> > ### **Technical innovation in methodology**
> >
> > We indeed do not claim to provide new methods for inserting memory into LLMs and SORT is indeed simple to construct, which allows to easily create new datasets as the need arises. Our contributions are a task to evaluate a form of memory that has so far been overlooked in the evaluation of LLMs. We hope our work will be useful to guide technical innovation of new memory architectures and we also look forward to more benchmarks that test episodic memory in LLMs as we define it here in new ways.
> >
> > ### **Do models know the book?**
> >
> > Most books on project gutenberg have a wikipedia article as well as summaries about them online that could have found its way into the training data of models. Our baseline results suggest that this is not a problem, i.e. abstractive knowledge about the books is not enough to order segments of text from the books with high accuracy.
> >
> >
> > ### **Omission of GPT-4 for memory-insertion**
> >
> > We know very little about closed-source models, i.e. their architecture, size, training, post-training is completely unknown. As we have also pointed out to reviewer 7fun, we believe that closed-source models are best used to show the top level of performance that a state-of-the-art model can reach on a task. For our in-context memory experiments, open models already demonstrated strong results.
> >
> > We hope that our clarifications on our contributions and the addition of a definition in the manuscript let you reconsider your evaluation of our work.
> >
> > 2/2

---

> > > ### Comment · Reviewer_kx1o · 2024-11-27
> > >
> > > Thank you for your response and clarifications (sorry, I missed your answer notification).
> > > While I appreciate your detailed explanation of the definition and scope of your work, I still have some reservations about the framing, scope and claims of the paper.
> > >
> > > Your new definition of episodic memory in LLMs, while structured, still seems to considerably narrow the rich concept of episodic memory as understood in cognitive science. The simple ordering of text segments captures only a minimal aspect of what constitutes true episodic memory, which involves the integration of spatial, temporal, contextual, and experiential details into coherent event representations. Calling this episodic memory is in itself an overclaim. I of course acknowledge your point about focusing on document processing for LLMs. Perhaps framing this more specifically as "sequential order memory" or "textual sequence memory" would be more accurate and avoid overclaiming.
> > >
> > > Your response about reported versus encountered events didn't fully address my concern. The ability to comprehend and remember reported episodic events (whether in literature, history, or personal accounts) is crucial for both humans and AI systems. This aspect deserves more exploration beyond simple ordering tasks ("AI is supposed to recall and master the episodic events of mankind").
> > >
> > > That said, I really appreciate the direction you're taking. I think it is a valuable first step in evaluating temporal memory in LLMs. The human evaluation is also valuable (perhaps the biggest contribution of this current work). If you want to keep the work as it is, I would encourage you to position it as an important but initial exploration of one specific aspect of memory in LLMs, rather than as an assessment of episodic memory.

---

> > > > ### Author Response · Authors · 2024-11-29
> > > >
> > > > Thank you for your response! We hope to further clarify our response with a follow-up.
> > > >
> > > > SORT is an initial evaluation task for episodic memory and not a complete assessment of all that ever should be evaluated about episodic memory in models. However there is a reason why temporal order memory is studied in psychology in relation to episodic memory and likewise we think that it presents a good starting point for evaluating episodic memory in LLMs.
> > > >
> > > > ### **Generality of our definition of episodic memory**
> > > >
> > > > Our definition, in particular criterion (3) is not constrained to temporal order relations between segments/events/items in a sequence. To make this more clear, we now rephrase this criterion in our revision of the paper. Other types of relations can also fall under this definition (see our new Appendix A for a more detailed discussion of our definition).
> > > >
> > > > ### **Is the reported order of events an important part of episodic memory?**
> > > >
> > > > In some sense, episodic memory can be about more “reported” aspects of temporal context binding, e.g. when positional markers and landmarks have to be used as temporal context to remember the spatial location (where did something happen?), or when reading an email, memory of the sender address is remembered and bound to the content of the email (who wrote the email?). While these are also important memory capabilities, we want to make a point that it’s the **encountered** order that is characteristic for episodic memory:
> > > >
> > > > **Example of a detective encountering reports about a crime**:
> > > > >A detective (whose episodic memory we want to discuss here) is interrogating 4 suspects about a crime. The suspects are reporting fragments of what happened during the crime, and the detective is reconstructing the events of what happened based on these reports.
> > > >
> > > > We would consider the contents of the detective’s reconstructed understanding of the crime scene to be a semantic aspect of memory, rather than episodic. Characteristically, episodic aspects of the reports here would concern how the detective remembers *who* of the suspects told *which part* of the story, and *when* they told this (in relation to other things that happened or were stated during or before the interrogation).
> > > >
> > > > ### **SORT as a partial rather than complete assessment of episodic memory**
> > > >
> > > > You are right in noting that SORT is a *partial* assessment of episodic memory, an initial benchmark task for this neglected form of memory, and we do not intend to claim otherwise in the paper. However we believe that temporal order memory is an integral aspect of episodic memory (hence its prominence in studies of episodic memory in humans) and that – *unless a model has been specifically designed only for sequence order recall* – high performance on SORT should be expected to correlate with higher performance on other aspects of episodic memory. We would include an empirical analysis of how different aspects of episodic memory are linked to the ability to recall sequence order, but SORT presents the first evaluation task for episodic memory, i.e. it is just a starting point for a new line of future research. We believe that the partial assessment of episodic memory in SORT will not limit the impact our work can have and instead hope that it will inspire the community to develop more evaluation tasks to test additional aspects of episodic memory.
> > > >
> > > > ### **Changes regarding the scope of the paper**
> > > >
> > > > To address your concerns, we have made several changes in our revision.
> > > >
> > > > 1. To make it more clear that SORT is not a complete assessment of episodic memory but just a first step, we updated the paper’s abstract, introduction and discussion. We would also consider changing the title to “Towards Assessing Episodic Memory in LLMs with Sequence Order Recall Tasks”, if this would be a convincing change for you. We wish to make it clear that we hope that SORT will be one of many future tasks that explicitly evaluate episodic memory in LLMs.
> > > >
> > > >
> > > > 2. We revised point (3) in our definition of episodic memory to highlight its generality for other aspects of episodic memory and have added an elaboration on this in Appendix A.
> > > >
> > > >
> > > > We think our contributions with SORT as an initial benchmark for a previously neglected form of memory will be important for the community and hope that you can reassess your evaluation of our work.

---

> > > > > ### Comment · Reviewer_kx1o · 2024-12-01
> > > > >
> > > > > Thank you for your detailed responses and attempted revision. While I appreciate the direction you're taking and the valuable contribution of the human evaluation study, I must maintain significant concerns about the paper's claims regarding episodic memory. Let me try to articulate them below.
> > > > >
> > > > > 1. Fundamental Nature of Episodic Memory
> > > > >
> > > > > First, here's "where I come from" in a bit more details. Episodic memory, as established by Tulving's seminal work and decades of subsequent research, is characterized by several key features that I miss in this work.
> > > > > - During *encoding*, a subject experiences a scene of the world. During *recall*, a partial piece of that scene like a smell or visual object (called a cue) is presented, leading to the recall (mental time travel) of the rest of the details of the encoded "scene".
> > > > > - Episodic memory involves hence the ability to reconstruct rich, multidimensional past experiences from such partial cues
> > > > > - It integrates the what/where/when information into coherent episodes, organized in time and space, forming a coherent cognitive map...
> > > > > - This context-dependent retrieval is further governed by mechanisms like encodingy specificity where the closer the "match" to the original context of encoding, the easier the retrieval etc...
> > > > >
> > > > > This has been supported by decades of research, that relate to the above traits, *in both humans and animals*.
> > > > > For humans, I'm thinking about autobiographical memory interviews that probe human's episodic memory, work on amnesia patients showing intact semantic memory but the inability to recall the details of events.. I'm thinking also about animal studies (e.g. Morris Water Maze, contextual fear conditioning), where rodents can encode complex episodes that they can retrieve thanks to partial environmental cues (leading to appropriate behaviors like correct navigation in spatial memory, or freezing in case of fear memory etc).
> > > > >
> > > > > So when we recall episodic memories, it's not just about the sequence of events and their order, but the layout of the room, the weather outside, the conversations, and the emotional context. This is fundamentally different from remembering that "baking happened before eating" in a sequence of kitchen-related events.
> > > > >
> > > > > 2. Core Issues with SORT as an Episodic Memory Task
> > > > >
> > > > > The SORT task, while well-designed for its actual purpose, tests a much narrower capability (as you actually mention), expressed in my terms:
> > > > > - Determining which of two text segments appeared earlier in a sequence
> > > > > - Testing awareness of linear ordering
> > > > > - Checking positional information tracking
> > > > >
> > > > > Now, *your proposed definition of "episodic memory in LLMs" appears (to me) to have been constructed to accommodate SORT rather than capturing the essence of episodic memory*. While I appreciate the attempt to adapt the concept for artificial systems, the current definition strips away the fundamental characteristics that make episodic memory distinct from other forms of memory. *I'm even worried that your definition misleads readers given how far it is from actual episodic memory in humans and animals (as described above)*. You can say that this is your new definition of episodic memory for AI. I even believe that AIs' episodic memory might need to be revisited differently (see some input in point 4 below), but this paper is may be not the best place to do so or argue about this (see my last point 6 below).
> > > > >
> > > > > 3. Fundamental Issues with Memory Types and Implementation
> > > > >
> > > > > Next, your discussion of parametric versus in-context memory reveals a significant conceptual flaw (to me). You argue that certain implementations "disqualify" as episodic memory, but you do so based on technical limitations (e.g., in-context memory's inability to handle arbitrarily long sequences, or parametric memory's poor SORT performance). This creates problematic technical distinctions that miss the fundamental nature of episodic memory.
> > > > >
> > > > > Whether a memory system qualifies as "episodic" should be determined by its functional characteristics - specifically its ability to encode and reconstruct rich experiences from partial cues - not by implementation details or current technical limitations. The constraints you describe (sequence length limits, SORT performance) are temporary technical limitations of current LLMs, not defining characteristics of episodic memory systems. *Future architectures might (I think will) implement genuine episodic memory through both parametric memory, and in-context mechanisms alike*.

---

> > > > > > ### Comment · Reviewer_kx1o · 2024-12-01
> > > > > >
> > > > > > Furthermore, the one-shot exposure definition you use is also arbitrary. Not all declarative memory is "one shot", frequency of exposure and saliency of the events play a role in ecoding performance of declarative memory. This is about the efficiency of remembering and has nothing to do with whether it is episodic or not.
> > > > > > *When I "rehearse" a particular insignificant event in my life, I end up remembering it (e.g. I'm trying to do so right now, until i'm sure i'll remember this point tomorrow); the fact that I rehearse this event to remember it, does not make it less episodic.*
> > > > > >
> > > > > > 4. The Nature of LLM "Memory": what do we want LLMs to remember?
> > > > > >
> > > > > > Finally, there's also a deeper issue regarding the type of memory we're discussing in LLMs, for LLMs. Current language models are primarily trained to learn and process humanity's collective reported experiences - what we might call "reported episodic memory" rather than "lived episodic memory". This is fundamentally different from biological episodic memory systems that encode their direct experiences (with the exception of humans that encode also reported events thanks to language).
> > > > > >
> > > > > > When a human reads about historical events or fictional narratives, they're processing reported episodes. Their episodic memory might include the experience of reading the text (where they were, how they felt), in addition to the reported events themselves.
> > > > > >
> > > > > > **An open question for episodic memory in LLMs is *what do we want them to recall?*, *reported episodic events"?* (what happened before what in the story we're giving the LLM in the sequence) or *"experienced" episodic events*? (what sequence of text came first).
> > > > > >
> > > > > > This is an important consideration that must be tackled when making the link between SORT and episodic memory. This leads me to the last final point.
> > > > > >
> > > > > > 5. Path Forward
> > > > > >
> > > > > > I’m afraid that talking about episodic memory in this paper could be both not suitable and misleading*, and the proposed revision did not address this concern, far from it. I actually don't understand why the authors didn't simply frame their paper as what it actually does, instead of the overstatements and overcomplication with episodic memory: you simply test sequence order recall in humans and LLMs.
> > > > > > My advise, moving forward, would be to get rid of all the arguments about episodic memory, as they don't add anything to the work (and might even confuse both, those who are familiar and those who are not familiar with episodic memory).
> > > > > > I think also that it's wiser to rethink your claims in the discussion, to focus on functional capabilities rather than current technical limitations.
> > > > > >
> > > > > > *If you're interested in elaborating the connection to episodic memory (i think that you started, that it's interesting, but that the analysis can be made deeper), why not writing a separate position paper that deeply develops the relationship between sequence ordering and episodic memory in AI? This paper does not seem to me the right place to do it.
> > > > > >
> > > > > > I am sorry that I cannot be more positive. I am certain that the next version of this work will be stronger, more faithful to what the paper actually achieves, and hence less misleading and more valuable to the community.

---

> > > > > > > ### Author Response · Authors · 2024-12-04
> > > > > > >
> > > > > > > > “Fundamental Nature of Episodic Memory”
> > > > > > >
> > > > > > > The definition you refer to is indeed rooted in the human literature. However, translating this concept to computational models, where notions of space and time are less easily defined, presents inherent challenges. The field of computational models of episodic memory offers a rich body of work that often emphasizes one-shot or few-shot learning as a central characteristic of episodic memory (e.g., [1,2,3,4]).
> > > > > > >
> > > > > > > Our proposed definition aims to operationalize the concept of episodic memory specifically for testing in LLMs. Notably, the single-shot criterion (2) does not imply that multiple exposures necessarily transform episodic memory into a non-episodic memory. Instead, it asserts that episodic memory can be established after a single exposure. This is a critical property—without it, no unique episode could ever be remembered, as each episode is inherently lived only once.
> > > > > > > We believe that criterion (3), which specifies that episodic memory *“contains information about relations between parts (e.g., encountered events/items, including more abstract items) within that sequence”* (p.1), is sufficiently broad to encompass diverse types of relations, including those represented in cognitive maps for various purposes that we do not evaluate here (refer to our references in Appendix A.1 and the general response).
> > > > > > >
> > > > > > > While the example you mention—“*remembering that ‘baking happened before eating’ in a sequence of kitchen-related events*”—is indeed not an instance of episodic memory, it pertains more to common-sense temporal order inference or reasoning. Within the scope of our definition, criterion (1) does not apply to this example. That said, episodic memories can certainly contain non-unique aspects of ordering; however, SORT does not focus on these aspects, as reflected in the low baseline performance.
> > > > > > >
> > > > > > > > “Core Issues with SORT as an Episodic Memory Task”
> > > > > > >
> > > > > > > You note that "*your proposed definition of 'episodic memory in LLMs' appears (to me) to have been constructed to accommodate SORT rather than capturing the essence of episodic memory*" and suggest that "*the current definition strips away the fundamental characteristics that make episodic memory distinct from other forms of memory.*"
> > > > > > > We want to clarify that the definition we provide is not arbitrary. As noted in our general response and in section A.1 of the appendix, we have referenced prior work to substantiate each criterion. The definition is specifically designed to operationalize the concept of episodic memory for application to LLMs. Importantly, the criteria we propose should be interpreted as necessary conditions for episodic memory, rather than as a comprehensive account that encapsulates all facets of human episodic memory.
> > > > > > > In light of your comment, we would like to emphasize again that criterion (3) accommodates a broad range of relationships that can be considered aspects of episodic memory. This flexibility is supported by the references cited in Appendix A.1, which illustrate the variety of relational structures that episodic memory can include.
> > > > > > >
> > > > > > > > “Fundamental Issues with Memory Types and Implementation”
> > > > > > >
> > > > > > > We appreciate your feedback on the limitations of in-context memory. To clarify, the limitation we highlight is not merely a technical constraint in the classical sense of the term. Episodic memory, as a form of long-term memory, requires support for arbitrarily long contexts. If a method cannot accommodate this, we would not consider it capable of enabling long-term, lasting episodic memories.
> > > > > > >
> > > > > > > You raise an important question: "*What do we want LLMs to recall—reported episodic events (what happened before what in the story we're giving the LLM in the sequence) or experienced episodic events (what sequence of text came first)?*"
> > > > > > > The concept of episodic memory inherently pertains to "*experienced episodic events,*" irrespective of the perspective from which we examine it. Notably, experienced episodic events can include reports of episodes as part of their content, and this is indeed frequently reflected in the training data of LLMs.
> > > > > > >
> > > > > > > References:
> > > > > > >
> > > > > > > [1] O'Reilly, R. C., Bhattacharyya, R., Howard, M. D., & Ketz, N. (2014). Complementary learning systems. Cognitive Science, 38(6)
> > > > > > >
> > > > > > > [2] O'Reilly, R. C., & Norman, K. A. (2002). Hippocampal and neocortical contributions to memory: Advances in the complementary learning systems framework. Trends in Cognitive Sciences, 6(12)
> > > > > > >
> > > > > > > [3] McClelland, J. L., McNaughton, B. L., & O'Reilly, R. C. (1995). Why there are complementary learning systems in the hippocampus and neocortex: Insights from the successes and failures of connectionist models of learning and memory. Psychological Review, 102(3)
> > > > > > >
> > > > > > > [4] Das, P., Chaudhury, S., Nelson, E., Melnyk, I., Swaminathan, S., Dai, S., Lozano, A., Kollias, G., Chenthamarakshan, V., Navrátil, J., Dan, S., & Chen, P.-Y. (2024). Larimar: Large language models with episodic memory control. ICML 2024

---

### Official Review · Reviewer_GiyF · 2024-11-04

**Soundness:** 2
**Presentation:** 4
**Contribution:** 2
**Rating:** 3
**Confidence:** 4

**Summary:**

The authors propose a new dataset, called SORT, for assessing episodic memory in LLMs, centered around the task of recalling the correct order of two short book snippets. The authors report extensive evaluations of modern LLMs on the proposed dataset, studying both in-context and "parametric" memory (memory encoded in weights and acquired during training), as well as externally augmented memory options. They find that modern LLMs can use in-context memory quite well, but struggle when the relevant context is only presented during training.

**Strengths:**

The paper addresses a highly important topic of assessing memory capabilities in LLMs.

I appreciate the attention to statistical detail reporting, and the generally thorough reporting of the experiments.

The paper is well-written.

The inclusion of human baselines, especially with the clever involvement of Goodreads users, is a great addition to the paper.

**Weaknesses:**

** Significance and novelty **

I believe that the work, overall, is sufficiently novel. At the very least, it provides a unique and potentially useful dataset.
I'm however, very conflicted on the significance of this work.

First, the authors do not discuss whether the notion of Episodic memory in LLMs is valid at all. "Episodic memory" is not clearly defined in the paper, other than type of memory "which links memories to their contexts, such as the time and place they occurred".

If we take this definition, many general-knowledge questions can be seen as probing episodic memory. E.g. questions in the form of "what was the movie in which a <SCENE DESCRIPTION>?", and "Describe that scene in more detail", or "When was this movie created" which modern LLMs are well-equipped to answer.

Usually, in psychology, episodic memory is distinguished from declarative memory by being related to personal experiences as opposed to stated facts. This distinction is extremely moot in the case of LLMs. This seriously undermines the premise of the paper.

Moreover, the key psychological literature the authors cite actually refers to a specific subset of human episodic memory, namely temporal memory. This creates confusion. If accepted, the authors should consider changing "episodic" to "temporal" throughout the article. It's worth noting that preserving temporal order is not a necessary attribute of episodic memory. In fact, humans routinely mix up the order of their personally significant memories; a simple way to see that is to try to reliably order all of one's childhood memories.

Still, the problem is deeper than this mismatch in terms. With no clear distinction between "personal" and "stated" memories in LLMs, what is the importance of studying temporal memory in LLMs in the way that authors suggest? Why, and to what extent, do we equate personal experiences with the order in which a some information is presented in the data?

If this is a way to address the issue of correcting LLM memories - why isn't there a deeper discussion of alternative methods and of how exactly this dataset will benefit the effort. If this is about whether or not LLMs can remember the order of things presented in contiguous chunks in training data - we know very well that LLMs are very good at that, e.g. that common mathematical sequences can often be retrieved if included in the training data. If it's about in-context handling of long sequences - why isn't there a more thorough discussion of other datasets that look at long-context tasks?

** Quality of the experimental support **

When it comes to human evaluation, the experiment seems well-organized and adds a lot of value to the paper.

I am, however, very troubled by many fine-tuning results and experimental decisions.

*Data ratio
The ratio of the new data to instruction data is very strange: the authors keep 3500 instruction following prompts "to avoid catastrophic forgetting", when there are only 116 independent book samples. It's very likely that in these conditions, the models simply do not have enough incentive to learn book-specific information. Additionally, the instruction fine-tuning stage behind Llama and Mixtral models is much more extensive than the 3500 instruction samples. Therefore, most likely, doing full fine-tuning on such a restricted sample severely hurts model performance across the board, making results hard to interpret. As I mention later in this review, it's not entirely clear why the authors even try to "preserve the models’ ability to understand and follow the task instructions" instead of simply fine-tuning a classification head for their specific task.

*Unusual training stage order:
Instruction fine-tuning and raw language modeling stages are mixed. I don't know how it affected results, but again, it makes interpretation harder. Instead, the authors could have chosen a much simpler two-step process: First, a language modeling fine-tuning stage on the books of interest (main condition) or on irrelevant books (control). Second, transfer learning fine-tuning with a classification head, treating SORT as a binary classification task. This would remove the need to keep a bunch of instruction-tuning samples and would clarify the results substantially. I understand that not all types of evaluations can be performed on these models, but that would more definitively answer whether LLMs can remember temporal information from their training set.

*Concerning training dynamics:
Another issue is that according to figure 16, the log probability of the correct answer is dropping with each new epoch. This is deeply concerning, and suggests that something is fundamentally wrong with the environmental set-up.

*Apparent failure to overfit:
It is well-known that LLMs are extremely good at verbatim memorization, and can overfit to memorize extreme amount of information. I am absolutely certain that the LLMs considered in the paper can be fine-tuned to memorize 9 books verbatim. The most likely reason that we didn't see improved performance is, then, either 1) inadequate training setup 2) inadequate methods of probing/information extraction (e.g. the model has the memory but its zero-shot skills are so degraded that it can't use this memory). Perhaps the authors are interested not only in whether a given "episodic" memory is implicitly present in LLM's parameters, but also whether it can it be used via zero-shot prompting? But that seems like a substantially different research question.

The last quality issue, not related to fine-tuning, is that the dataset is destined to become obsolete. I.e. as the public domain books in question enter the training sets of new LLMs, much of the dataset's value will be lost, and the reported results won't be comparable to an attempted replication on newer LLMs. I don't think this is a disqualifying consideration, but it does negatively affect the significance and potential impact of this work.

Overall, these issues largely invalidate the conclusions on whether parametric memory in transformer can be seen as a form of episodic memory.

A minor note: I believe it'd be beneficial to the paper to add an extra classification baseline where a model is finetuned to predict which segment comes earlier in a book, without any context. It's the same as "Baseline" in the paper, but with fine-tuning. The reason it's important is that it could be that zero-shot instructions are not enough to extract this information from the model. Of course, since there are only 9 books in the dataset, it'd be necessary to either do cross-validation (fine-tune on 8 books, evaluate on 1).


** Summary **

Overall, I highly resonate with the importance of creating new datasets to thoroughly evaluate LLM memory capacity. I deeply appreciate the inclusion of a human-subject study and a thorough approach to evaluation. That being said, unfortunately, I can't recommend the work for acceptance.

The main issue is the positioning of this work in the broader context. What exactly is the problem we are trying to address? Is this a useful benchmark that should use to test memory capabilities of new models, or is it more of an investigation into the nature of episodic memory in transformers? I feel that the paper attempts to tackle both questions at once, but, as a result, does not sufficiently address either. Another substantial problem is the experimental support: I find that many of the fine-tuning experiments are not sufficiently clearly constructed and executed.

My true overall score for this submission is 3.5, or 4. I think that even if published as is, it will provide some positive value to the community. That being said, unfortunately, in its present form, it is not strong enough for ICLR.

**Questions:**

I am sorry I could not provide a more positive evaluation. Clearly, a lot of work went into this paper, and it pains me to recommend a rejection. I deeply hope that the authors can slightly re-focus the paper, making it more focused on a specific problem or question. And if the question is the nature of LLM memory as compared to human memory, a much deeper discussion of characteristics and properties of different types of human memory would be necessary. It's also important to clearly decide on the distinction between having a memory and being able to access it. For example, some model might be able to recite Hamlet, but be unable to answer questions about it. Should we treat such a model as having the memory of Hamlet or not? In any case, the fine-tuning experiments need to be improved, as right now they are, in my view, the weakest part of the paper. Again, I am sorry I can't provide a more positive assessment at the moment.

I hope the authors continue their work and I'd be happy to see a revised version of this paper published one day.

I will also, of course, read the other reviews and the rebuttal. I am not sure how likely it is, but I will be open to reevaluate my assessment if it turns out that I overlooked some important considerations.

Questions:

What is the criterion for having an episodic memory? If a model has it "implicitly" (see the Hamlet example above) -- does it count?

What was the reason behind trying to maintain instruction-answering capabilities of the LLMs instead of switching to a simple Supervised Fine-Tuning classification task to see if the information is internalized?

What is the main goal of the paper? Specifically, do the authors see it as more of an inquiry into the nature and characteristics of LLM memory, or is it primarily a benchmark to evaluate the newly released models' memory capacity?

Minor:
On line 162, the authors say that the segments are "preceded by X", which would mean providing the whole book. But this contradicts the actual experiments where only book samples are actually presented due to context length limitations.

** Extra sources **
If we treat this paper as primarily a useful benchmark for assessing trainsformers' memory capability, NarrativeQA (https://huggingface.co/datasets/deepmind/narrativeqa) and, even more so, NarrativeXL (https://arxiv.org/pdf/2305.13877), datasets propose a different approach to a similar problem of assessing long-term memory in transformers, and use a similar methodology. These and related works should probably be discussed to better define the role of the present contribution.

---

> ### Author Response · Authors · 2024-11-20
> **Episodic memory definition and new fine-tuning results**
>
> We highly appreciate your engagement with our work and your thoughtful comments and suggestions. Based on your comments, we have provided a clearer and more in-depth definition of episodic memory in LLMs and provide new experiments to affirm the results from the fine-tuning memory-insertion method.
>
> ### **Defining episodic memory in LLMs**
> We have updated the manuscript with a proposed definition of episodic memory in LLMs. This definition is heavily based on features of human episodic memory, but does not require exact equivalence to human memory. As stated briefly in the introduction, we believe that these features of episodic memory may be key to improving LLM performance on several cognitive tasks.
>
> > **Definition 1.1: Episodic Memory in LLMs**
> > Episodic memory refers to knowledge in a language model that:
> > 1) is specific and unique to a particular sequence;
> > 2) is acquired through a single exposure to that sequence (single-shot learning);
> > 3) contains information about the unique context of segments within that sequence (e.g.
> the temporal and general composition of events/items within the complete sequence);
> > 4) can still be retrieved when arbitrarily many tokens are processed in between the
> sequence’s encoding and its retrieval;
> > 5) has functional implications, meaning the knowledge can be used by the model to
> answer explicit queries.
>
> This definition does not include any reference to personal experience, as we agree that this does not make sense for LLMs. Instead, it focuses on the specificity of knowledge to a particular sequence, meaning that learning from a distribution of *different* sequences does not lead to knowledge about unique features of that sequence.
>
> ### **New finetuning experiment with overfitting**
> We believe the decreasing log-probability of the correct answer for our fine-tuned models (Llama3-8b-Instruct) is due to the changes in instruction-following behavior that come with fine-tuning for multiple epochs on the same limited amount of data (Allen-Zhu & Li, 2023). This memory-insertion has not failed to insert memory of the segments, because we show that the perplexity of the segments decreases.
>
> To highlight the importance of instruction-following and the insufficiency of overfitting, we now additionally fine-tune a model with the LM objective on (A) 300 and (B) 120 repetitions of the book-text while training on only (A) 30k and (B) 300k unique instruction data samples with batch-size set to 24. For (A), this is close to a ratio of 1:1 between book-chunks and unrelated instruction-data.
>
> To verify the presence of implicit memory of the text in the resulting model, we evaluate the perplexity for the book texts, showing a substantial decrease in perplexity from 9.9 to 2.2 (A) and 2.3 (B). Instead of relying on the model's output layer, we trained a linear SORT-task readout with book-wise cross-validation (we fit a linear readout on SORT-samples of 8 books, then evaluate on SORT-samples from the remaining book). While the perplexity on The Murder of Roger Ackroyd decreased from 9.9 before finetuning to 2.2 (A) and 2.3 (B) after finetuning, our cross-validated logistic regression shows that performance is at chance level for all books. At the same time, in-context memory performance on SORT decreased to 52% in (A) and 82% in (B), highlighting the need for more instruction data to counter the detrimental effects of overfitting.
>
> ### **Relation between verbatim memorization (implicit memory) and episodic memory as tested with SORT**
>
> Following our definition of episodic memory, the requirement (5) for access to temporal context memory for question-answering would disqualify implicit temporal order memory, unless a model is additionally able to answer questions about its temporal context, requirement (3). Overfitting a model on a single text comes with catastrophic forgetting of instruction-following ability, which includes the ability to answer explicit questions. Besides the requirement (5) in our definition for an LLM to remain able to understand the task, it has been shown that LLMs memorize texts only if they are presented multiple times (Carlini et al. 2023, Figure 1b; Ovadia et al. 2024, p.8) - as you note, it is *common* mathematical equations that are remembered by models. The reliance on repetitions for verbatim memorization violates requirement (2) in our definition: episodic memory needs to be single-shot learned.
>
>
> ### **Instruction data and the formation of memory during fine-tuning**
> We agree that the need for instruction-data is a valid concern and a major problem with finetuning of already instruction-tuned models. For the insertion of semantic knowledge, it has been shown that a model's instruction-following ability gets harmed if no instruction data is given during fine-tuning (e.g. Allen-Zhu & Li, 2023), and we likewise found that not including enough instruction data severely harms SORT performance, *even when the relevant excerpt is given in-context*.
>
> 1/3

---

> > ### Author Response · Authors · 2024-11-20
> > **SORT as a benchmark of episodic memory**
> >
> > ### **SORT is a benchmark**
> >
> > We think of SORT as an evaluation task that should be used together with other benchmarks to get a more complete picture of new and existing models’ memory capabilities. SORT datasets are created synthetically, making it easy to create new datasets with longer sequences, without requiring human annotations. As you note, in the future it will be necessary to update datasets used for SORT, eventually making our initial dataset BookSORT obsolete. We provide code to quickly create new SORT datasets for new texts. Our primary aim is to provide a new, easily extendable resource that allows to evaluate how well episodic memory can be inserted into a model. We tested three common methods for inserting memory into LLMs. For memory-augmented models, entirely new methods for inserting memory could be evaluated.
> >
> > ### **Why complete task fine-tuning as an alternative to instruction-following is undesirable for a benchmark task**
> >
> > Given that we intend to use SORT as a benchmark of memory capabilities, fine-tuning the model on the task (or training a new binary read-out layer) instead of relying on its ability to follow instructions would be undesirable, as it would hide the downsides of inserting episodic memory via fine-tuning in any realistic scenario. Catastrophic forgetting would present a serious problem and thus, measures will need to be taken to preserve prior abilities, e.g. by including samples from the previous data distribution (i.e. instruction data) in the fine-tuning data. It might be interesting to use SORT not just as a benchmark and study existing architectures and insertion-methods more deeply, but that will be the investigation of future work.
> >
> >
> > ### **Order-preservation in episodic memory**
> > The task of temporal order judgments is a popular evaluation task used to study episodic memory in humans (Eichenbaum, 2013; Davachi & DuBrow, 2015). We trust in decades of research and base SORT on this task.
> >
> > ### **Relation to NarrativeXL/NarrativeQA**
> > NarrativeXL presents a large supervised training dataset to enhance the abilities of models to deal with long-form narratives and perform in reading comprehension and free recall tasks.
> >
> > *Given our understanding of episodic memory in LLMs, can reading comprehension serve as an episodic memory benchmark?*
> >
> > The authors of NarrativeXL note that questions can have a different answer when asked at different points in a book (there is a lot of ambiguity). They thus provide read-along questions to be asked while a model is processing a book in-context. This is contrary to our requirement (4). We want to test the ability with an arbitrary number of processed tokens in between the relevant sequence and the query.
> >
> > SORT avoids this type of ambiguity. Given that segments are unique in SORT, there is always a correct answer for any segment-pair.
> >
> > Some questions on reading-comprehension (e.g. the Ghostbusters II question shown in Figure 1 in NarrativeQA) are about single, isolated pieces of knowledge, which we do not consider to be suitable for testing episodic memory due to the irrelevance of broader temporal context (requirement 3). In other words, not every retrieval task tests episodic memory. In SORT, memory of contextual relations between a segment and its surrounding context with a potentially arbitrarily long horizon is crucial to what we test. For the retrieval of e.g. facts, surrounding context could be replaced with noise (as is done in needle-in-a-haystack tests), highlighting the decontextualized nature of such retrieval. With this said, we believe that NarrativeXL could be filtered in the future to extract a subset of comprehension questions that is suitable to test episodic memory in that it requires proper contextualization. We now mention these highly relevant datasets in our related work section. The relation between performance on SORT and other tasks that potentially test episodic memory is also an exciting idea for future work that we will add to our discussion.
> >
> >
> > 2/3

---

> ### Author Response · Authors · 2024-11-20
> **Other concerns, questions and references**
>
> ### **Extendability of BookSORT**
>
> The risk that BookSORT will no longer be useful in the future (because new models will be trained on these books) is realistic, however it does not present a problem for SORT, because we provide code that allows to easily create new versions of SORT-datasets as the need arises, which includes the possibility of using new proprietary texts for the purpose of internal testing.
>
> ### **Answers to your other questions**
>
> 1. Implicit memory would not qualify as episodic memory primarily because it requires multiple presentations or content that is not unique to the sequence, e.g. common mathematical equations (see req. 1 and 2 in our definition of EM in LLMs).
> 2. We need to maintain instruction-following capabilities because SORT requires it and req. 5 in our definition requires explicit querying to work.
> 3. Our primary goal is to provide a resource to the community that can be used to behaviorally assess episodic memory (as defined in the paper now) in LLMs to obtain a more complete picture of memory in language models. We intend it to be used not in isolation but in addition to other evaluations of LLM capabilities. SORT is always an evaluation of a model together with a particular memory-insertion method. New memory-architectures may afford new methods for inserting memory that would not be possible with standard transformer-based models.
>
> We thank you again for your comments as they've helped us improve our paper, and hope we could clarify the main points you've raised. We’d appreciate the opportunity to engage in a discussion if some points have remained unclear.
>
> ### *References*
>
> Allen-Zhu, Z. and Li, Y. Physics of Language Models: Part 3.1, Knowledge Storage and Extraction, 2023, https://arxiv.org/pdf/2309.14316
>
> Carlini, N., Ippolito, D., Jagielski, M., Lee, K., Tramer, F., & Zhang, C.. Quantifying memorization across neural language models. 2022, https://arxiv.org/abs/2202.07646
>
> Ovadia, O., Brief, M., Mishaeli, M., & Elisha, O. . Fine-Tuning or retrieval? Comparing knowledge injection in llms. 2023, https://arxiv.org/abs/2312.05934
>
> Eichenbaum, H. Memory on time. Trends in cognitive sciences, 17(2):81–88, 2013.
>
> Davachi, L. & DuBrow, S. How the hippocampus preserves order: the role of prediction and
> context. Trends in cognitive sciences, 19(2):92–99, 2015.
>
> 3/3

---

> ### Comment · Reviewer_GiyF · 2024-11-27
> **Re: rebuttal**
>
> I appreciate the thoughtful and detailed answers to my questions. Unfortunately, I can not raise my score since some of my key concerns remain un-addressed.
>
> 1) My main concern is still with the significance and focus of this work.
>
> The authors provide a revised definition of what they mean by "Episodic Memory" in LLMs with five sub-criteria. It does help to be more specific in the object of study, but it is still, unfortunately, problematic.
>
> For example, the criterion 2 ("knowledge is acquired through a single exposure to that sequence (single-shot learning")) seems arbitrary.
>
> What is the meaning of restricting episodic memories to a single exposure to a training item & doing only one gradient step based on this piece of knowledge? This has no exact parallel in humans; if anything, it goes vaguely contrary to how episodic memory works in humans and animals. Episodic memories are not memorized "instantly", rather, they are consolidated during sleep. In other words, human brains likely do multiple equivalents of "gradient updates" to acquire a single episodic memory.
>
> Criterion 5 ("[episodic memory knowledge] has functional implications, meaning the knowledge can be used by the model to
> answer explicit queries") is also, in my view, arbitrarily tailored to the work already done. The authors are testing LLMs on something they were not trained to do (recalling sequential information about training items). It is natural that "out of the box," these models will struggle on the task, so restricting ourselves to zero-shot and few-shot queries seems strange. It is very likely that these models have the "episodic memories" we are interested in, and are simply struggling to explicate them since it's a dramatically non-standard task. The use of various probing techniques is highly natural and popular in such cases. I strongly disagree that the only correct way to assess the model's knowledge is to ask it about it through prompting.
>
> 2) Experimental quality.
>
> I appreciate additional discussion of the (lack of) overfitting issue. Training a linear readout is a step in the right direction (if the main focus of the paper is assessing episodic memory in LLMs). At the same time, with only 8 books in the training set, I am afraid that there are not enough unique data points to train the model or make strong conclusions. If the authors dropped the "single exposure" requirement, it would dramatically broaden the scope of data available for this. For example, they could have trained the model to explicate implicit temporal knowledge on thousands of openly available books, and then tested it on the new ones.
>
> **** Conclusion ****
>
> At the end of the day, the authors are testing the models on something these models are not trained to do (accessing temporal order of specific sub-items in the training set). The methodology requires fine-tuning on a mixed dataset and involves a tricky balance of a) training data we are interested in with b) instruction fine-tuning data we include to avoid catastrophic forgetting.
>
> As a result, applying the proposed benchmark to assess episodic memory produces hard to interpret results (e.g. if the model fails at the task it might mean that it did not acquire the memory, but it might also mean that it lost the ability to answer queries, etc.).
>
> Such a hard-to apply benchmark would have been justified if it evaluated something crucial. Unfortunately, the connection to human episodic memory remains tenuous and the connection to practical LLM applications remains even less clear. I don't believe that the authors make a strong enough case for the relevance of their proposed evaluation.
>
> All in all, unfortunately, I believe that the authors aim to achieve two goals at once: on the one hand, they draw a parallel between human episodic memory and one particular way of assessing memory in LLMs, on the other hand, they propose a benchmark that could be useful more broadly (for evaluating in-context memory capacity). Unfortunately, I believe that due to this attempted breadth, the paper lacks sufficient depth in both of these aspects.
>
> I genuinely believe that the paper has a potential for being expanded into two separate & successful publications.
>
> a) One pathway is to focus on the problem of long-context & in-context temporal memory in transformers. If the authors could show that SORT offers new insights not covered by existing evaluation protocols/benchmarks (needle in a haystack-like tasks with two needles seem a strong competitor), SORT could join the ranks of standard benchmarks in that area and benefit the community.
>
> b) Another is to dive deeper into the topic of episodic memory in transformers. In this case, the authors won't have to try to restrict their post fine-tuning analysis to basic prompting, and can provide a much deeper look into the problem, as well as a much deeper discussion of why "episodic memory," as they define it, is something we should be looking into.
>
> 1/2

---

> > ### Comment · Reviewer_GiyF · 2024-11-27
> > **re: rebuttal (2/2)**
> >
> > Again, I am sorry that I can not recommend this paper to acceptance. I understand that a lot of work has been put into it, but I must consider how/whether this work is likely to benefit the ML community. Unfortunately, right now, in my estimate, it does not provide sufficient depth and focus, largely because of how ambitious the undertaking is. I hope that my negative assessment does not discourage the authors and that they reframe and deepen their work, potentially splitting it into two publications.
> >
> > Right now, the attempt to assess something fundamentally new and human-psychology-related (episodic memory in LLMs) is in substantial conflict with the alternative goal of using this dataset to evaluate something obviously important and standard (in-context LLM memory). This "duality" is not disqualifying in and of itself, but it makes it much harder for the paper to deliver on both of these aspirations.
> >
> > Minor notes:
> >
> > On order preservation in human & animal episodic memory:
> > I also trust in decades of research in psychology. As studies suggest, episodic memory does not perfectly preserve global temporal order (https://psycnet.apa.org/fulltext/2012-09921-001.html , https://pmc.ncbi.nlm.nih.gov/articles/PMC3081724/), even though it usually does preserve more local temporal sequence. If the paper wants to go deeper into the "episodic memory" domain, it'd be important to discuss in more depth what should be considered one "episode" and why, and how it corresponds to different types/scales of human episodic memory.
> >
> > On the new definition:
> > Criterion 1 seems to be overlapping with criterion 3. It might be better to combine them for brevity. Generally, it's important to add a much more substantial discussion of the implications, similarities & differences between human episodic memory and LLM episodic memory, to justify why it is crucial to study this phenomenon.

---

> > > ### Author Response · Authors · 2024-11-29
> > > **Non-arbitrarity of our definition of episodic memory**
> > >
> > > ### **SORT is tailored to the definition of episodic memory, not the other way around**
> > >
> > > We are proposing a definition of episodic memory in LLMs that is based on what is known about episodic memory in psychology and neuroscience. This definition aims to differentiate it from other forms of memory, like semantic memory or procedural (implicit) memory, to highlight a current shortcoming of the field, and we attempt to fill this gap with a new and easily extendable evaluation task. We present an explanation of each criterion of our proposed definition below, and have added an elaboration in *Appendix A*, where we provide further references from neuroscience and psychology on these properties of episodic memory for each of the criteria.
> > >
> > > **Criteria (1) & (2)** are defining characteristics of episodic memory that highlight how episodic memory allows to *quickly* form memory traces based on a single exposure to a specific sequence. This does in fact not involve repetitive presentation of the sequence. Repetition has even been shown to harm episodic memory in some cases (Peterson et al., 2012). If there was a way to fine-tune an LLM *without repeatedly encoding the raw text* but instead relying on internal representations of the text, this could provide multiple gradient updates based on the same sequence. In the brain, such consolidation indeed plays an important role in stabilizing episodic memories  – it is however not a feature of the initial formation of the memory, where only a single exposure is given.
> > >
> > > **Criterion (3)**, now rephrased for better clarity on its generality, is different from criteria (1) in that it expresses how episodic memory captures structural and relational aspects of a sequence. The criterion is included to differentiate it from memory of facts where the temporal context in which the facts are stated and how they relate does not matter.
> > >
> > > **Criterion (4)** expresses that episodic memories have potential to last for an arbitrary amount of time. This indeed links episodic memory evaluation to long-context evaluation, but it is only one aspect of this form of memory, as there are other forms of long-term memory that are more about knowledge of facts.
> > >
> > > **Criterion (5)** is motivated by the generally explicit/declarative nature of episodic memory: content of episodic memories can be used for explicit reasoning and communication. Without this requirement, an assessment of episodic memory in LLMs would not be informative about whether a given model can use contextual order information that may have previously been encoded in some form (e.g. into its weights). Successful linear probing of representations (as an alternative to our explicit task) would not mean that the LLM can use this information to perform any task where it would be relevant. The fact that a linear probing model can distinguish between two categories based on representations does not mean that the identified information is *functionally* relevant (Belinkov, 2022). Including this criterion makes sure that we require models to be able to actually retrieve temporal order information associated with parts of a document that it has memory of.
> > >
> > >
> > > 1/2

---

> > > > ### Author Response · Authors · 2024-11-29
> > > > **Main point of the paper**
> > > >
> > > > ### **Training models on SORT would limit its value as an initial evaluation task of episodic memory**
> > > > In this paper, we introduce a new evaluation task that can be used to test the emergence of episodic memory (an aspect of memory that has been neglected) for any combination of model and memory-insertion method. Proposing novel methods to enable LLMs to perform the task is not part of our paper. As previously stated, we introduce SORT as a benchmark to evaluate whether a feature of episodic memory can be explicitly recalled.
> > > >
> > > > A training scheme in which LLMs are first trained on book texts and then fine-tuned to recall the order of segments in those texts, based on parametric memory, would not just constitute benchmarking of a model and a memory-insertion method – it would be an attempt to *improve/enable* parametric episodic memory in models. More importantly however, for such a model it would be less clear whether the emergent ability to retrieve order information might indicate general episodic memory capabilities. Since no other benchmark consistently tests episodic memory in LLMs, there would be no way to interpret positive results in the broader context of episodic memory, which would still be given if a model is not specifically designed/optimized just for our task, but good performance emerges. We made a related point in the discussion section on whether order-preserving RAG could qualify as episodic memory.
> > > >
> > > > **Needle-In-A-Haystack Analogy:**
> > > > For the sake of further illustration of this point, let’s assume the needle-in-a-haystack task was the first and only existing benchmark for long-context evaluation. Training a model with limited long-context abilities to retrieve needles in the presence of distractor documents would not be expected to improve its general long-context abilities, even if the model ends up performing well on a needle-in-a-haystack test set. Since in this scenario there would be no other long-context tasks, there’d be no way to interpret the results and arrive at the conclusion that the emergent long-context abilities are actually very limited, unlike suggested by the test. Using a needle-in-a-haystack test on models that are *not explicitly trained for it* provides a better assessment of their general long-context abilities, since the tested ability is an emergent byproduct and not a specifically engineered one.
> > > >
> > > > This being said, we agree that better understanding how and where in a model temporal context information can be stored in model parameters is interesting and important to study - it is just not the point of our work.
> > > >
> > > > ### **Main purpose of this paper**
> > > > The focus of our paper lies in providing an evaluation task for the community that can be used to assess how well (model, memory-insertion-method) combinations support a form of memory that has previously been neglected in LLM evaluations. We investigate different memory-insertion methods primarily to highlight that (contrary to current mainstream focus of research) SORT is not just intended as a long-context evaluation but that it should be used to discover alternative methods to long-context prompting (due to its obvious disadvantages). We believe that presenting SORT to the community in this way will aid in the development of new memory-insertion methods that go beyond injecting semantic knowledge.
> > > >
> > > > ### **Relevance of the proposed evaluation**
> > > > There are several problems in machine learning that may be addressed if models could bind the temporal context to memories. As we state in the introduction of our paper, more human-like episodic memory may improve models' continual learning and adaptation to shifting data distributions. Binding of temporal context to memory content can also potentially address issues with source attribution, and therefore help in dealing with hallucinations, by providing information about where and in which context a memory was acquired.
> > > >
> > > > ### **Summary of our response**
> > > > We hope that you can reconsider your evaluation of our work in the light of the points we've made:
> > > >
> > > >
> > > > 1. Our definition of episodic memory is grounded in literature and is not arbitrary.
> > > >
> > > > 2. The main purpose of this paper is to provide a new evaluation task that allows to test a neglected form of memory in LLMs, which has potential to guide the development of future memory-insertion architectures.
> > > >
> > > > 3. Training a model specifically on our task is not a good idea if the goal is to test broader emergent episodic memory capabilities, and it would go beyond the scope of this paper, which is to evaluate existing models and methods, to show that SORT is not intended as just another long-context evaluation task.
> > > >
> > > >
> > > > **References**
> > > >
> > > > Belinkov, Y. (2022). "Probing Classifiers: Promises, Shortcomings, and Advances." Computational Linguistics, 48(1), 207–219
> > > >
> > > > Peterson, D. J., & Mulligan, N. W. (2012). A negative effect of repetition in episodic memory. Journal of Experimental Psychology: Learning, Memory, and Cognition, 38(6), 1786–1791.
> > > >
> > > > 2/2

---

### Author Response · Authors · 2024-12-04

We thank the reviewers for engaging in a discussion with us, and have found it valuable to articulate our work more clearly. Unfortunately, we were not able to come to a joint understanding, with two reviewers supporting acceptance [Reviewers 7fun and yhUH] and two rejection [Reviewers GiyF and kx1o].

The central point of disagreement lies in what constitutes episodic memory in LLMs. This is an emerging area of research, so it is natural to have differing perspectives. Our work proposes a definition of episodic memory in LLMs that we believe is helpful for its assessment:
>Episodic memory refers to knowledge in a language model that:
>1. is specific and unique to a particular sequence [1,2];
>2. can be acquired through a single exposure to that sequence (single-shot learning) [3,4,5,6,7,8];
>3. contains information about relations between parts (e.g. encountered events/items, incl. more abstract items) within that sequence [1,9,10,11];
>4. can still be retrieved when arbitrarily many tokens are processed in between the sequence’s encoding and its retrieval [12,13,14,15];
>5. has functional implications, meaning the knowledge can be used by the model to answer explicit queries [14,15].

Reviewers GiyF and kx1o are concerned that our definition is overly restrictive for different reasons. We respectfully stand by our definition, as we believe it captures *necessary conditions of episodic memory* that:

1. draw on insights from human and animal research as well as a rich body of computational models of episodic memory [1-15]; and
2. provide a practical framework for operationalizing this type of memory in LLMs for testing purposes.

We clarify that these criteria form an initial definition, which does not  fully encapsulate the richness of episodic memory and will be refined by future work and ongoing scientific discourse. Our primary goal in defining episodic memory in this way is to operationalize the concept specifically for LLMs, enabling the evaluation of a previously untested aspect of memory in these models. By providing this initial evaluation framework, we aim to equip the research community with a practical and flexible tool to assess and advance the memory capabilities of language models. To achieve this, we:
- propose Sequence Order Recall Tasks (SORT) as a way to test a new aspect of memory in language models, distinct from memory for context-invariant facts;

- show how SORT can be used to evaluate different methods to insert memory;

- provide a first-of-its-kind human dataset (N=155), serving as a human performance reference for SORT;

- provide open SORT dataset generation and evaluation code to enable evaluation on new, long-context data as LLM memory capabilities continue to be updated

We believe our work will provide a useful resource and inspire useful discussions, just as it has done here already.

#### References
[1] J. Sugar and M. Moser. Episodic memory: Neuronal codes for what, where, and when. Hippocampus, 2019.

[2] L. Colgin, E. Moser, and M. Moser. Understanding memory through hippocampal remapping. Trends in neurosciences, 2008.

[3] Z. Liao and A. Losonczy. Learning, fast and slow: Single- and many-shot learning in the hippocampus. Annual Review of Neuroscience, 2024.

[4] B. Schwartz and S. Evans. Episodic memory in primates. American Journal of Primatology: Official Journal of the American Society of Primatologists, 2001.

[5] R. O'Reilly., R. Bhattacharyya, M. Howard, and N. Ketz. Complementary learning systems. Cognitive Science, 2014.

[6] R. O'Reilly and K. Norman. Hippocampal and neocortical contributions to memory: Advances in the complementary learning systems framework. Trends in Cognitive Sciences, 2002.

[7] J. McClelland, B. McNaughton, and R. O'Reilly. Why there are complementary learning systems in the hippocampus and neocortex: Insights from the successes and failures of connectionist models of learning and memory. Psychological Review, 1995.

[8] P. Das, S. Chaudhury, E. Nelson, et al. Larimar: Large language models with episodic memory control. ICML, 2024.

[9] J. O’Keefe and L. Nadel. The Hippocampus as a Cognitive Map. Oxford: Clarendon Press, 1978.

[10] H. Eichenbaum and N. Cohen. Can we reconcile the declarative memory and spatial navigation views on hippocampal function? Neuron, 2014.

[11] H. Eichenbaum. The hippocampus as a cognitive map . . . of social space. Neuron, 2015.

[12] A. Mayes and N. Roberts. Theories of episodic memory. Philosophical Transactions of the Royal Society of London. Series B: Biological Sciences, 2001.

[13] M. Conway. Sensory–perceptual episodic memory and its context: Autobiographical memory. Philosophical Transactions of the Royal Society of London. Series B: Biological Sciences, 2001.

[14] L. Squire and S. Zola. Structure and function of declarative and nondeclarative memory systems. PNAS, 1996.

[15] R. Hampton and B. Schwartz. Episodic memory in nonhumans: what, and where, is when? Current opinion in neurobiology, 2004.

---

### Meta-Review · Area_Chair_LQUT · 2024-12-08

**Metareview:**

This submission assesses the abilities of a set of language models (LMs) to perform "sequence-order recall", that is, recalling the order of elements of a previously seen sequence. The authors describe the capacity that performs this task in LMs "episodic memory", and give an informal definition of this capacity in LMs. The target of the submission is to relate the "episodic memory" capacity in LMs and humans by testing LMs and humans in matched tasks of "sequence-order recall". The comparison demonstrates that in-context sequence-order recall in LMs is superhuman but degrades with context length, suggesting that it is operating more like "working memory" and in conflict with recent work that connects in-context learning and episodic memory. This is an interesting discussion point. Other forms of LM memory (fine-tuning and RAG) are tested to see their performance on sequence-order recall tasks (SORT). The submission shortly concludes that these forms of memory are also unlike human/animal episodic memory.

The evaluation of "sequence-order recall" to provide evidence for "episodic-memory" like capabilities is novel and interesting. However, the relationship between episodic memory in humans and the sequence-order recall tasks presented here is muddy: As Reviewer GiyF, the capability to recall sequence order is not exactly episodic memory in humans. Moreover, the same control variables are not in use across the human and LM studies (namely, context lengths in Figs. 3, 4 are not varied for humans), making it difficult to conclude that the same mechanisms are at play to substantiate a direct comparison. A revision of the submission should appropriately scope the discussion of episodic memory in LMs as directly related to a memory capacity in humans alongside the narrowness of the evaluation task.

**Additional Comments On Reviewer Discussion:**

During the rebuttal period, Reviewers GiyF and kx1o strongly critiqued the paper's characterization of episodic memory, arguing that sequence order recall captures only a minimal aspect of episodic memory in humans/animals. While Reviewers yhUH and 7fun were more positive about the paper's novel evaluation framework and human baselines, the core disagreement remained about whether SORT adequately assesses episodic memory, suggesting the paper would be stronger if reframed around sequence order memory specifically. The authors attempted to address these concerns by providing a revised definition and additional experiments, but were ultimately unable to fully convince skeptical reviewers about the broader episodic memory framing.

---

### Decision · Program_Chairs · 2025-01-22

Reject